

**Variations of Arctic winter ozone from the LIMS Level 3 dataset**

Ellis Remsberg[1], Murali Natarajan[1], and Ernest Hilsenrath[2]

[1]Science Directorate, NASA Langley Research Center, 21 Langley Blvd, Mail Stop 401B, Hampton, VA 23681, USA

[2]Fellow at Joint Center for Earth System Technology, University of Maryland at Baltimore County, 1000 Hilltop Circle, Baltimore, MD 21250, USA

Correspondence to: Ellis Remsberg (ellis.e.remsberg@nasa.gov)

(for submission to Atmospheric Measurement Techniques Journal)

February, 2022

**Abstract**

The Nimbus 7 limb infrared monitor of the stratosphere (LIMS) instrument operated from
October 25, 1978, through May 28, 1979. Its Version 6 (V6) profiles and their Level 3 or zonal
Fourier coefficient products have been characterized and archived in 2008 and in 2011,
respectively. This paper focuses on the value and use of daily ozone maps from Level 3, based
on a gridding of its zonal coefficients. We present maps of V6 ozone on pressure surfaces and
compare them with several rocket-borne chemiluminescent ozone measurements that extend into
the lower mesosphere. We illustrate how the synoptic maps of V6 ozone and temperature are an
important aid in interpreting satellite limb-infrared emission versus local measurements,
especially when they occur during dynamically active periods of northern hemisphere winter. A
map sequence spanning the minor stratospheric warmings of late January and early February
characterizes the evolution of a low ozone pocket (LOP) at that time. We also present time
series of the wintertime tertiary ozone maximum and its associated zonally varying temperatures
in the upper mesosphere. These examples provide guidance to researchers for further
exploratory analyses of the daily maps of middle atmosphere ozone from LIMS.

## 1 Introduction and objectives

The historic Nimbus 7 Limb Infrared Monitor of the Stratosphere (LIMS) experiment provided
data on middle atmosphere ozone from October 25, 1978, through May 28, 1979, for scientific
analysis and for comparisons with atmospheric models (Gille and Russell, 1984). Ozone is an
excellent tracer of stratospheric transport in the high latitude stratosphere. As an early example,
Leovy et al. (1985) showed how daily maps of the LIMS ozone fields correlate well with
geopotential height (GPH) fields on the 10-hPa pressure surface. They also reported on the
rapidly changing effects of wave activity on ozone, which led to a better understanding of
stratospheric transport processes within models. Hitchman et al. (1989) also analyzed the
temperature fields from LIMS and reported on Arctic observations of an elevated stratopause in
late autumn to early winter that they associated with momentum forcings from gravity waves.

Current research focuses on the 3-dimensional character of ozone in the upper stratosphere and
mesosphere, based on more recent satellite datasets.  Several studies consider how temperature
and ozone vary in association with sudden stratospheric warming (SSW) events (Smith et al.,
2009; de la Camara et al., 2018; Kim et al., 2020; Shams et al., 2021).  Manney et al. (1995) and
Harvey et al. (2008) describe the development of low ozone pockets (LOPs) in the region of the
Aleutian anticyclone during winter.  Siskind et al. (2005; 2021) explain the occurrence of a
mesospheric cooling associated with SSWs and the role of gravity waves for modeling ozone in
the upper mesosphere, respectively.  Chandran et al. (2013) provide a climatology of the Arctic
elevated stratopause, and Sofieva et al. (2021) analyze for regional trends in stratospheric ozone.
Smith et al. (2011; 2018) report on monthly changes of the tertiary ozone maximum at high
latitudes of the upper mesosphere during winter.

The LIMS (Level 2) profiles were retrieved with an improved Version 6 (V6) algorithm.  They
were archived in 2008 and include ozone, temperature, and GPH that extend from 316 hPa to
~0.01 hPa.  Co-located V6 profiles of water vapor ($H_2O$), nitric acid vapor ($HNO_3$), and nitrogen
dioxide ($NO_2$) extend through the stratosphere.  Lieberman et al. (2004) analyzed the V6
temperature profiles and found evidence for non-migrating tides in the mesosphere, due to the
interaction of the diurnal tide and planetary zonal-wave 1, especially in late January 1979.  Holt
et al. (2010) analyzed the descent of V6 $NO_2$ from the lower mesosphere to within the polar
stratospheric vortex, where it interacts with ozone.  Remsberg et al. (2013) assimilated V6 ozone
profiles in a reanalysis model and gained improved estimates of column ozone, especially in
Arctic winter.  Such reanalysis studies assimilate temperature and ozone profiles within a model
framework.  However, the models only approximate the effects of small-scale variations, so it is
also useful to consider observed variations of the LIMS parameters without resort to a model.
Keep in mind that smaller-scale atmospheric variations also contribute to the analyzed
intermediate and large-scale fields from V6.  This paper further explores several instances of
those larger-scale variations of Arctic ozone, temperature, and GPH.

The SPARC Data Initiative (SPARC-DI) includes monthly zonal averages of V6 ozone up to the
0.1-hPa level (see Tegtmeier et al., 2013; SPARC, 2017; and Remsberg et al., 2021).  In Section
2 we show January zonal averages of V6 ozone and temperature profiles that extend even higher
or to near the mesopause.  The V6 Level 3 (map) product provides a 3-dimensional context for
those zonal mean data.  Daily V6 maps are also an aid in interpretating individual V6 profiles
versus correlative data, especially during dynamically disturbed periods.  Specifically, in Section
3 we compare several nighttime V6 ozone profiles with those obtained with a rocket-borne
chemiluminescent technique (Hilsenrath et al., 1980).  Those profile comparisons are for
December 15 and for January 27 and 28, when the temperature and ozone fields were affected by
planetary wave forcings.  There is a corresponding cooling and variations of ozone in the winter
lower mesosphere associated with the warming in the upper stratosphere.  Section 4 presents
variations of ozone and GPH at northern extratropical latitudes during the minor SSW events of
late January and early February 1979, as a complement to the more comprehensive findings of
Harvey et al. (2008) on the occurrence of LOPs within anticyclones determined from satellite
solar occultation data.  Section 5 considers the variability of the tertiary ozone maximum in the
upper mesosphere during that same period, as an adjunct to monthly zonal average values
reported by Smith et al. (2018).  Section 6 notes that the maps of V6 ozone contain more details
about gradients in temperature and ozone and cautions users about occasional, pseudo-ozone
features in the tropical lowermost stratosphere.  Section 7 concludes that the V6 Level 3 product
is an important resource for studies of the effects of transport and chemistry on Arctic ozone.

## 97     2    Characteristics of V6 Level 3 data

*2.1 LIMS measurements and analyses*
Nimbus 7 was in a near-polar orbit, and LIMS made measurements at ~1 pm local time along its
ascending (A or south-to-north) orbital segments and at ~11 pm on its descending (D or north-to-
south) segments.  A-D time differences are of the order of 10 hours at most latitudes because
LIMS viewed the atmosphere 146.5° clockwise of the spacecraft velocity vector, as seen from
above.  The A-D differences narrow from 10 to about 6 hours from 60°N to 80°N, due to the
orbital geometry of Nimbus 7.  The V6 processing algorithm accounts for low-frequency
spacecraft motions that affect the LIMS view of the horizon.  As a result, its measured radiance
profiles are well registered in pressure-altitude (Remsberg et al., 2004).  Retrieved V6 ozone,
temperature, and GPH profiles extend from 316 hPa to ~0.01 hPa and have a vertical point
spacing of ~0.88 km with an altitude resolution of ~3.7 km.  Retrieved profile pairs are spaced
every 144 km along the orbital track or at every 1.3°, but closer together at the high, turn-around
latitudes of the orbital viewing geometry (Remsberg et al., 1990).  LIMS made measurements
with a duty cycle of about 11 days on and 1 day off over its planned observing lifetime.  The
LIMS algorithms (Remsberg et al., 2007) do not account for non-local thermodynamic
equilibrium (NLTE) effects in ozone (Solomon et al., 1986; Mlynczak and Drayson, 1990) and
in $CO_2$ (Edwards et al., 1996; Manuilova et al., 1998), so there are positive biases in the retrieved
V6 ozone throughout the mesosphere during daylight.  However, the V6 nighttime ozone is more
nearly free of NLTE effects below about the 0.05-hPa level, except at times of SSWs (see e.g.,
Funke et al., 2012).

A sequential-estimation (SE) algorithm was used to generate daily, zonal Fourier coefficients
(zonal mean and up to six cosine and sine values or 6-zonal wavenumbers) for Level 3 at every
2° of latitude and at up to 28 vertical levels (Remsberg and Lingenfelser, 2010).  The V6 SE
algorithm uses better estimates of data uncertainty and its zonal wave coefficients have a
memory of ~2.5 days, or about half that of the SE algorithm used by Remsberg et al. (1990).
The SE analysis is insensitive to the very few large, unscreened ozone profiles values found in
the lower stratosphere, as noted in Remsberg et al. (2013, their Fig. 1a).  The SE algorithm
combines the coefficients from both the separate A and D orbital segments and effectively
interpolates the profile data in time to provide a continuous, 216-day set of daily zonal
coefficients versus pressure-altitude at 1200Z for each of the retrieved LIMS parameters.

*2.2 Monthly average V6 data*

One can generate monthly average distributions from the daily Level 3 files of temperature,
GPH, and species (ozone, $H_2O$, $HNO_3$, and $NO_2$); zonal averages for the V6 species were
supplied to SPARC-DI (SPARC, 2017; Hegglin et al., 2021).  Tegtmeier et al. (2013) compared
the V6 monthly ozone distributions with ones from other satellite-based, limb sensors and
reported good agreement throughout the stratosphere.  Although the species cross sections for
SPARC (2017) extend only up to the 0.1-hPa level (~64 km), V6 average ozone extends higher
or to about 0.015 hPa (~75 km). Figure 1 shows the latitude-pressure cross section for January
from just the descending (D) orbital profiles, which avoids the larger NLTE biases that affect
daytime ozone in the mesosphere. Stratospheric ozone mixing ratios in Fig. 1 have largest
values at about 10 hPa near the Equator (> 9.2 ppmv), and they decrease sharply above and
below that level. Maximum mixing ratios for the middle to high latitudes occur between 3 to 5
hPa, due to the larger zenith angles and longer paths of the ultraviolet light for production of
atmospheric ozone. There is a nighttime ozone minimum of ~1.2 ppmv across most latitudes of
the middle mesosphere. A tertiary ozone maximum is present in the upper mesosphere near the
winter day/night terminator zone in the LIMS measurements for January (at about 67°N), in
accordance with the interpretation of Marsh et al. (2001). The location (~0.02 hPa) and
magnitude (~3.5 ppmv) of the NH maximum are somewhat higher and larger than those reported
by Smith et al. (2018, their Fig. 4) from more recent satellite datasets. Although the V6 ozone
poleward of ~55°S is also from descending orbital profiles, it corresponds to daylight conditions
at the high southern latitudes in January. Thus, the decrease of mesospheric V6 ozone at 0.1 hPa
and poleward of 55°S indicates merely a change from night to day values and agrees with
findings of Lopez-Puertas et al. (2018). On the other hand, the rather large ozone values in the
upper mesosphere at about 45°S are not found in other data sets and are not expected from
theory. We consider that ozone anomaly further in Section 6.

Radiances from two 15-μm $CO_2$ channels are used for retrievals of V6 temperature versus
pressure or T(p), and they are free of NLTE effects below about the 0.05-hPa level (~70 km)
(Lopez-Puertas and Taylor, 2001). To first order, the V6 T(p) retrievals account for the effects
of horizontal temperature gradients in the stratosphere (Remsberg et al., 2004). Single profile
root-sum-squared (or RSS) errors for T(p) vary from 1 K at 10 hPa to ~2.5 K in the upper
mesosphere, but they do not include possible temperature gradient errors. RSS error from T(p) is
the primary source of bias error for ozone, growing to about 16% in the middle mesosphere
(Remsberg et al., 2021, Table 1). Random errors become large for single ozone profiles in the
upper mesosphere. As a complement to the V6 ozone of Fig. 1, we show the descending
(~nighttime) V6 T(p) distribution for January in Figure 2, which extends to near the 0.01-hPa
level. The large-scale features of the T(p) distribution compare well with climatological values
from the late 1970s (Fleming et al., 1990), having a maximum value of about 285 K at the SH
high latitude stratopause and minimum values of < 200 K at the tropical tropopause and near the
SH summer mesopause.  There is also some elevation of the Arctic zonal-average stratopause.

Figure 3 shows the monthly-average, zonal (wave) standard deviations (SD) about daily zonal
means of the combined-mode (A+D) V6 ozone for January, where the SD values are derived
from the zonal-wave amplitudes of V6 Level 3.  There are relatively small SD values at low
latitudes from 7 to 10 hPa; it is assumed that they are a result of smaller-amplitude Kelvin and
Rossby-gravity waves.  Effects of more vigorous, planetary wave activity are most apparent at
high northern latitudes of the stratosphere during winter.  Gravity waves also contribute to SD in
the uppermost mesosphere (Siskind et al., 2021).  Ozone shows little zonal variation in the SH
upper stratosphere of Fig. 3, due to constraints on the upward propagation of planetary waves
through the summer zonal easterlies (Andrews et al., 1987).  SD values near the tropical
tropopause are due mostly to residual effects of emissions from thin cirrus and represent spurious
ozone variations (see Section 6).

**3.   V6 comparisons with rocket-borne chemiluminescent ozone measurements**
In this section we consider V6 comparisons with three nighttime, rocket-borne chemiluminescent
ozone soundings of Hilsenrath (1980)—one at White Sands, NM, (32.4°N, 253.5°E) on
December 15, 1978, and two more at Poker Flat, AK, (65.1°N, 212.5°E) on the successive days
of January 27 and 28, 1979.  The estimated total, rocket ozone error is 14% (precision plus
accuracy), according to Hilsenrath and Kirschner (1980).

Ozone comparisons for December 15 are in Figure 4 (top); we plot every other V6 profile and
those four profiles have spacings of 2.6° in latitude.  The short-dashed V6 profile is at 29.2°, and
the long-dashed profile is at 37.2°.  The solid curve is the V6 profile at 31.8° (at 0611Z) or
closest to the rocket sounding from White Sands (at 0541Z).  Horizontal bars on the profiles are
estimates of ozone error; they overlap between V6 and rocket, except in the upper stratosphere.
LIMS ozone is larger than rocket ozone in through the upper stratosphere.  The corresponding
V6 ozone map at 4.6 hPa in Fig. 4 (bottom) reveals an ozone maximum just south of White
Sands (WS—blue dot), along the descending orbital segment of the satellite at (6°N, 265°E—
white dot) or viewing in the NNW direction toward White Sands.  Note that while zonal
variations in the map are from a gridding of the Level 3 coefficients (2° latitude and 5.625°
longitude), there is no smoothing of the gridded field in the meridional direction; there is good
continuity across latitudes, nonetheless.  The rocket profile is a local measurement and has a
vertical resolution that ranges from 1.5 km at 60 km to 0.1 km at 20 km; the nearby V6 profiles
have a lower vertical resolution of ~3.7 km and are an average over the finite horizontal length
(~300 km or ~3° latitude) of the LIMS tangent layer.  There is an ozone maximum along the
LIMS view path just to the south of White Sands, which may account for the profile differences.
We also note that the ozone field of two days earlier has the region of sharp gradients positioned
over White Sands with ozone at only 8 ppmv.  Thus, an ozone field that varies in both space and
time can lead to additional uncertainties for comparisons of the localized rocket and limb-
viewing satellite profiles in Fig. 4.

Because V6 ozone is obtained from retrievals of the measured V6 ozone radiance profiles, the
LIMS retrieved temperature profile must be representative of the atmospheric state for the
forward model of ozone radiance.  Figure 5 (top) shows the corresponding temperature
comparisons between V6 and a separate rocket Datasonde instrument.  Agreement between them
is very good throughout the upper stratosphere, indicating that the temperature variations are
well determined along the LIMS view path for the forward radiance calculations of V6 ozone
and that the retrieved V6 ozone should be nearly unaffected by temperature bias error.  The map
of V6 temperature (Fig. 5—bottom) shows zonal variations on December 15, although their
meridional gradients are relatively weak above White Sands.  Conversely, the ozone profiles
agree well near 0.68 hPa in Fig. 4, where there are apparent biases between the T(p) profiles.
There are significant horizontal gradients near White Sands in the maps of T(p) at 0.68 hPa, but
not in ozone (not shown).  In fact, the V6 ozone field at that level has a nearly constant value,
and ozone is less sensitive (by half) to changes in T(p) at 0.68 hPa than at 4.6 hPa (Remsberg et
al., 2007).  Co-location is more important for the V6 versus rocket comparisons of T(p) than of
ozone in the lower mesosphere.

The two comparisons above Poker Flat, AK, occurred at the time of a stratospheric, zonal wave-
1 warming event.  Leovy et al. (1985) provide a detailed discussion of the advective changes for
ozone in the middle stratosphere during January 1979.  Figure 6 (top) shows three V6 ozone
profiles from along an ascending orbital segment on January 27.  The LIMS instrument was
viewing from its satellite location (80.7°N, 113°E) at 2204Z, and the rocket ozone launch was
two hours earlier or at 2005Z at a solar zenith angle of 84° or near the terminator; there is good
agreement of the structure between them, even in the mesosphere.  A second rocket launch
followed at 0833Z of January 28 (Hilsenrath, 1980).  Since the separate V6/rocket ozone and
T(p) comparisons are similar for the two days, Fig. 6 contains results for January 27 only.  The
rocket sounding recorded two ozone maxima, one near 15 hPa and another at about 0.6 hPa.  The
ozone maximum at about 15 hPa is primarily due to advection of ozone of higher mixing ratios
from lower latitudes just prior to the warming event.  The local maximum at 0.6 hPa was
unexpected, based on findings from a larger set of rocket ozone soundings.  There is a relative
minimum for both V6 and rocket ozone through the upper stratosphere, although V6 ozone is
larger.  The map of V6 ozone at 4.6 hPa in Fig. 6 (bottom) indicates that the rocket measurement
occurs at the center of the minimum, whereas the V6 profiles are averages across it.  The ozone
profiles in Fig. 6 (top) indicate the relative minimum in a low-ozone pocket (LOP) that extends
from about 7 hPa to 2 hPa.

Figure 7 (top) shows the V6 temperature profile comparisons; T(p) from the Datasonde has more
vertical structure, as expected from a localized measurement.  V6 T(p) values reach a maximum
of order 250 K at about 3 to 4 hPa.  They agree reasonably with the Datasonde values, given that
there is significant horizontal structure in the temperature field surrounding Poker Flat.  The
apparent V6 minus Datasonde bias of order 5 K at 3 hPa ought to lead to a V6 minus rocket
ozone bias of -40%, according to error estimates for retrieved V6 ozone.  However, Fig. 7
(bottom left) indicates that LIMS was viewing Poker Flat across an area of higher temperatures,
such that it is likely that there is a spatial mismatch for V6 and Datasonde T(p) values.  The
much smaller and positive ozone differences in Fig. 6 support that likelihood.  There may also be
co-location differences between the rocket temperature and ozone soundings in this instance.

Figure 7 also shows a map of NH GPH at 4.6 hPa on January 27 for comparison with the ozone
map in Fig. 6.  Lowest ozone values are in the polar vortex, where the GPH field is asymmetric
about the Pole.  A second, low value of ozone is associated with the anticyclone over the
Alaskan sector.  One can determine horizontal winds from gradients of GPH on the 4.6-hPa
surface and thereby estimate the transport of ozone to first order.  Qualitatively, the direction and
strength of the large-scale transport follows from the character of the cyclonic and anticyclonic
features on the GPH map.  The large-scale cyclonic circulation about the vortex transports air
from middle latitudes to across the Pole on January 27.  The vortex region has low ozone and is
relatively cold, whereas stratospheric temperatures over Alaska show a maximum (the SSW),
and the rocket profile above Poker Flat, AK, was near the center of the anticyclone and in the
region of relatively low ozone (or LOP).

Ozone is an approximate tracer of transport processes and reveals dramatic changes with altitude
associated with this SSW event, even through the winter lower mesosphere.  As an example,
Figure S1 (in Supplemental Materials) shows a concurrent cooling at 0.46 hPa above the Alaskan
anticyclone on January 27, where the co-located ozone field exhibits a local maximum.  There is
also a major temperature increase above the polar stratospheric vortex over northern Europe at
0.46 hPa, or where ozone values remain low.  In summary, Figs. 4 through 7 and S1 indicate the
utility of daily maps from LIMS for analyses of the ozone fields during dynamically disturbed
conditions.

**4.  Variation of a low ozone pocket (LOP) from LIMS Level 3**
The polar vortex on January 27 was located over northern Europe and Asia; it was centered off
the Pole because of effects of large-scale, planetary waves in the development of the SSW
(Andrews et al., 1987, Chapter 6).  In this section, we show sequences of polar plots of both
stratospheric GPH and ozone for February 1979.  Manney et al. (1995) and Harvey et al. (2004,
2008) provide comprehensive analyses about the occurrence of polar anticyclones and their
associated LOPs from studies of GPH and ozone fields from several different satellites.  They
determined the extent and character of the polar vortex based on meteorological data from the
UK Met Office or as obtained from relatively low vertical resolution radiance profiles from
operational, nadir temperature sounders.  The V6 GPH profiles are derived from and have the
same vertical resolution as the T(p) profiles.  Manney et al. (1995) showed that water vapor is a
useful tracer of the meridional transport of air, and the V6 $H_2O$ fields at 6.8 and 10 hPa indicate
that low latitude air was transported to the region of the LOP in late January.  But the V6 $H_2O$
fields are noisy at 4.6 hPa (not shown).  Even so, the V6 Level 3 ozone, T(p), and GPH data
offer useful details about the occurrence of LOPs in the upper stratosphere.

Harvey et al. (2004) reported that LOPs occur nominally at about the 5-hPa level.  Accordingly,
the three panels of Figure 8 show three daily NH maps of V6 GPH from February 3 to February
17 at 4.6 hPa; each successive map is spaced one week from the previous one.  This sequence
shows that both the vortex and anticyclone weaken during the three weeks following January 27
at this level.  The vortex re-centers on the Pole by February 17, and the anticyclone is nearly
absent at 4.6 hPa following the two minor warming events.  The map sequence of GPH indicates
that there were significant changes in the horizontal transport of ozone in late January/early
February.  The corresponding three panels of ozone in Figure 9 show the further evolution of
ozone, following that of January 27 (in Fig. 6).  Even though the anticyclone had weakened
during the first week, there was a deepening of the LOP from January 27 to February 3 and a
filling of it thereafter.

Was there some chemical loss of ozone from January 27 to February 3 in the region of the LOP?
Morris et al. (1998) and Nair et al. (1998) conducted model calculations to show how that could
happen.  Ozone reactions are affected by changes with latitude of solar insolation, temperature,
and loss via $NO_x$.  Nair et al. (1998) reported on the effect of a decrease in the production of
ozone for the development of LOPs, as air parcels in the middle stratosphere move from low to
high latitudes or to higher solar zenith angles in winter.  Remsberg et al. (2018) analyzed air
parcel trajectories that included chemistry, and they showed that there was some loss of ozone in
the middle stratosphere, due to reactions with $NO_x$. However, Holt et al. (2012) analyzed V6
$NO_2$ in the winter polar vortex, and they did not find enhanced values at 4.6 hPa due to energetic
particle precipitation (EPP) by late January.

Figure 10 (left) is a map of the V6 descending orbital (nighttime) $NO_2$ for January 27 at 4.6 hPa.
Based on the corresponding map of GPH in Fig. 7, one can trace the horizontal advection of high
$NO_2$ toward higher latitudes and toward the polar vortex as well as the advection of low $NO_2$ out
of the vortex and about the anticyclone. Fig. 10 (right) is a map of $HNO_3$ at 4.6 hPa, and it
shows a weak, relative maximum above the anticyclone.

The formation of the LOP at 5hPa near Poker Flat region in January is studied with the help of
photochemical calculations along a trajectory. A full description of the trajectory and
photochemical models is given in Remsberg et al (2018). For this study, we generated backward
trajectory starting at 5 hPa and 212° E and 64°N. The starting time of the back trajectory is 0900
Z on January 28,1979. This is close to local time of LIMS descending mode observation of
around 11:00 pm on January 27. Figure 11 shows the back trajectory with the day numbers
illustrating the progress of the air parcel. Between January 22 and 28, the air parcel remains
confined within a small region at high latitude. Results of a time dependent photochemical
calculation conducted along this trajectory in the forward direction are shown in Figure 12 as a
function of time. The model initialization uses the mixing ratios of $O_3$, $NO_2$, and $HNO_3$ from
LIMS descending mode observations. Figure 12 shows that the parcel was at 3 hPa on January
15 and descended steadily to 5 hPa by late January. The decrease in ozone during the last 8 days
is mainly caused by the daytime odd oxygen loss due to the catalytic cycle involving $NO_x$.
Production of odd oxygen is minimized since the air parcel is confined to high latitudes. Large
diurnal variations in $NO_2$ and a small increase in $HNO_3$ are as expected for this pressure level.
The LOP formation is a result of the interplay between transport and photochemistry in the high
latitude upper stratosphere in the winter.

### 5. Variations of the tertiary ozone maximum

Smith et al. (2018) describe the changing monthly, zonally averaged character of the wintertime tertiary ozone maximum of the polar upper mesosphere. They point out that the low latitude edge of the tertiary ozone maximum is where $HO_x$ radicals and the chemical loss of ozone due to reactions with them are reduced. V6 ozone radiance profiles have low signal-to-noise in the upper mesosphere; the precision estimate is 0.32 ppmv for retrieved ozone profiles. We show a map in Figure 13 of the combined V6 ozone for December 15 at 0.022 hPa (~72 km), where its distribution in the subpolar region is based on fewer than 13 zonal coefficients because some profiles do not extend to that pressure altitude. The corresponding map of temperature is also in Fig. 13, and one can see that there is significant non-zonal structure in its field at the latitudes where ozone is enhanced. While both V6 ozone and temperature are not highly accurate due to NLTE effects in the upper mesosphere, their maps reveal significant relative spatial structures indicating advective transport and its likely effects on ozone.

Figures S2 and S3 in the Supplemental Materials show additional panels at 0.022 hPa of ozone and temperature, respectively, for January 13, February 10, and March 1. Elevated values of ozone occur at higher latitudes on February 10 and March 1 than on December 15 and January 13, which is consistent with the more northward position of the terminator away from winter solstice and the consequent effects for the chemical loss of ozone. The temperature fields are also perturbed on January 13 and February 10, but they are more nearly zonal by March 1. However, there are meridional gradients of temperature on all three days in the region of the tertiary ozone maximum. On January 13 there is also a well-defined mesospheric vortex in GPH (not shown), and the highest values of ozone correlate reasonably with it. The vortex is most disturbed and tertiary ozone maximum has largest values on February 10, perhaps in response to the upward propagation of wave activity following the minor SSW of late January.

Figure 14 shows time series of peak zonal mean ozone at 0.022 hPa and its latitude location for each week from November through mid-March. The separate time series are for peak ozone (bottom two series) and their latitude locations (top two). Dashed red curves represent zonal

mean results for combined (A+D) ozone; solid black curves are results for nighttime (D) only.
Blue horizontal lines represent the approximate latitude position of the terminator. Peak
nighttime ozone values are based on just the 'zonal mean' and the cosine and sine coefficients
for waves 1 and 2 because not all profiles reach to the 0.022-hPa level. Peak ozone occurs at
lower latitudes (~65°N) in December, increasing to ~75°N in early November and early February
and to near 80°N by early March. The latitude time series of peak ozone values is reasonably
coincident with the changing location of the terminator. Peak combined (A+D) ozone increases
slowly from a minimum of 2.2 ppmv in November to 3.6 ppmv in late February and March.
Descending (or nighttime only) ozone varies from 3.3 ppmv in November, to ~4.5 ppmv in
January, to a maximum of 6.3 ppmv in mid-February, and then declining to 3.5 ppmv by mid-
March, although the time series shows rather large variations. Those maximum V6 values are
larger than reported by Lopez-Puertas et al. (2018, their Fig. 15), perhaps due to biases from V6
T(p) and/or ozone at 0.022 hPa.

The increase of V6 ozone during winter in Fig. 14 disagrees with that of Smith et al. (2018), who
found decreasing ozone by February. They reported that, in most years, there is a slow descent
of relatively dry air into the vortex region in the upper mesosphere during late autumn and early
winter, and that the reduction in water vapor implies that there are fewer $HO_x$ radicals for the
destruction of ozone near the terminator zone, leading to accumulations of ozone. However,
there were two minor warmings and associated lower mesospheric cooling events during late
January and early February 1979 (Hitchman et al., 1989). The enhanced V6 ozone of February
1979 follows those SSW events, and there are wave-driven disturbances and dissipation of their
energy in the upper mesosphere at that time (e.g., Siskind et al. 2005; Smith et al., 2009). One
can gain more details about the evolution of the tertiary ozone maximum in the winter of 1978-
79 from the daily maps of V6 ozone, T(p), and GPH (as in Figs. S2 and S3).

**6. Other aspects of V6 Level 3 ozone**
The combined (A+D) Level 3 coefficients are the basis for a gridding of daily synoptic maps at
1200Z of ozone and related parameters. The Level 3 product also contains coefficients from its
separate A and D profiles; their 'zonal mean' values correspond to the local time-of-day of their
respective measurements.  Remsberg et al. (2007) noted that maps from V6 reveal more details
about the variations of ozone.  In Figure S4 of the Supplemental Materials we compare a map of
V6 ozone at 10 hPa on 27 January with a similar map for V5 of Leovy et al. (1985).  The ozone
gradients are more pronounced with V6 than with V5 at both the subtropical and vortex edges of
the ozone field.   The V6 maps make use of all profiles along the orbit, and the SE mapping
algorithm was applied to them every 2° of latitude.  However, the tighter gradients were also
achieved with the V6 algorithm because it has a relaxation time (or memory) that is half that of
V5.  This means that the V6 maps are more representative of the rapidly changing atmospheric
ozone fields on that day.  Similar version differences are evident throughout winter, when the so-
called 'stratospheric surf zone' develops and expands (Leovy et al., 1985).

Significant exchanges of air and ozone occur from the extratropical stratosphere to the
troposphere in winter and spring (Gettelman et al., 2011).  There are large zonal variations about
the daily zonal means of ozone in the Arctic region of the lower stratosphere in Fig. 3.  There are
similar variations in GPH (and derived winds) and in zonal wave activity that lead to ozone
transport.  Zonal variations are resolved in the daily ozone maps down to the 146-hPa level.
Notably, Shepherd et al. (2014) integrated the V6 monthly zonal mean ozone above the
tropopause and subtracted it from observed total ozone, as part of their assessment of long-term
trends of tropospheric ozone from models.  Their determination of extratropical tropospheric
ozone based on LIMS agrees with that obtained from other ozone datasets.

There is also a relative excess of SD ozone values in Fig. 3 centered at 68 hPa at tropical
latitudes, and similar anomalies occur in other LIMS months (not shown).  As an example,
Figure 15 shows a map of V6 ozone at 68 hPa (~18 km) on December 15, to give more insight
about the source of the tropical variations.  Ozone mixing ratio values in Fig. 15 are of order 2 to
3 ppmv at high latitudes, becoming much smaller in the subtropics.  However, there is also an
unexpected, high value of 2 to 3 ppmv at about 15°N, 150°E.  Limb measurements in the ozone
channel include radiance effects from cirrus particles that can occur along the tangent view path,
although the retrieved ozone mixing ratio profiles were screened of those effects to first order
(Remsberg et al., 2007). Even so, we note that ozone is easily affected by any excess radiance
because of highly non-linear effects for retrievals of ozone in the lower stratosphere. It is very
likely that the anomalous ozone at 68 hPa is a result of residual effects from subvisible cirrus,
which is nearly ubiquitous over the western tropical Pacific region (see SPARC, 2006, Fig. 1.8).
While individual V6 ozone profiles may include such spurious features in the tropics, the Level 3
ozone product at 68 hPa is affected mainly when there is an organized convection and outflow of
air that persists for several days. The adjacent map of ozone at 46 hPa appears unperturbed in
that region (not shown), and tropical ozone at 100 hPa approaches zero. There are much smaller
anomalies in maps of nitric acid, as its mixing ratio retrieval is very nearly linear. Anomalies are
also not so apparent in maps of V6 $H_2O$ at 68 hPa because the cloud screening algorithm for $H_2O$
accounts for the larger vertical field-of-view and extent in altitude for measurements in the water
vapor channel of LIMS. Thus, one must be mindful that the Level 3 product may indicate
excess, but spurious ozone at 68 hPa in the tropics.

Finally, in our earlier description of descending orbital ozone in Fig. 1, we noted that there are
anomalously high values near 45°S in the upper mesosphere. Fig. S5 of the Supplemental
Materials shows distributions of temperature and ozone from descending orbital measurements at
0.032 hPa on January 15. Zonal wave activity is very weak in both. The descending orbital
views of the tangent layer at 45°S are located just in front of the region of large temperature
gradients at 40°S. Although we showed in Section 3 that taking account of temperature
gradients is important for accurate retrievals of ozone, such gradients were not employed for
calculations in the mesosphere of tangent layer radiance in the V6 algorithms. If such
temperature biases persist through the upper mesosphere, they will also affect the registration of
the observed ozone radiance versus pressure at those altitudes. Therefore, we judge that it is
very likely that the enhanced ozone mixing ratios near 45°S in Fig. 1 are an artifact because the
associated, retrieved tangent path temperatures are not weighted properly, are too cold, and do
not account for enough of the observed ozone radiance. Ozone mixing ratio anomalies are not
apparent along the ascending orbital segment because their LIMS views are in a near-zonal
direction.

## 7. Conclusions

This report provides guidance to researchers for their use of the LIMS V6 Level 3 product and for their generation of daily gridded distributions of its temperature, ozone, and GPH on pressure surfaces. $H_2O$, $NO_2$, and $HNO_3$ are also available for the stratosphere from the Level 3 product. The V6 dataset represents an early baseline for considering possible changes in the middle atmosphere from 1979 to today and into the future. LIMS made measurements at a time when stratospheric effects from volcanoes were minimal and when catalytic effects of chlorine on ozone were relatively small. Accordingly, Stolarski et al. (2012) found small, but significant changes in the distribution of upper stratospheric ozone for recent decades compared with 1978-1979. The LIMS measurements were taken near solar maximum and when atmospheric concentrations of the greenhouse gases (GHG), $CO_2$, $CH_4$, and CFCs, were smaller than today. Middle atmosphere T(p) distributions were warmer in 1978-1979.

The LIMS measurements in the winter Arctic region occurred when there was a lot of wave activity for the transport and mixing of ozone. As a result, ozone varied dramatically in winter, particularly during times of stratospheric warming events. There was a so-called Canadian warming in early December 1978, two minor SSW events in late January and early February, and a final warming in late February 1979. We showed V6 comparisons with temperature and ozone profile data obtained using rocket borne Datasonde and chemiluminescent instruments, and we pointed out how an examination of changes in their nearby fields is valuable for the interpretation and validation of V6 profiles against those correlative measurements. The Level 3 dataset provides daily details on variations of ozone with latitude, longitude, and altitude, along with related variations in temperature, geopotential height, $NO_2$, and $HNO_3$. We noted also that there are instances of spurious, excess ozone from the Level 3 coefficients at 68 hPa in the tropics but not in the extratropical stratosphere.

We displayed evidence of a low ozone pocket (LOP) and its chemical properties at 5 hPa above the Aleutian anticyclone during the minor SSW of late January, and we followed its evolution into mid-February. The V6 nighttime ozone is relatively accurate through the mesosphere in

Arctic winter.  We provided time series of the wintertime, tertiary ozone maximum of the upper
mesosphere from V6 data.  Its ozone reached maximum values in February, perhaps as a
response to enhanced wave activity in the mesosphere following several SSW events.  Together
with V6 maps of T(p) and GPH, one may explore further the daily evolution of that ozone
maximum throughout the NH winter of 1978-1979.

**Data Availability**

The LIMS V6 Level 3 product is at the NASA EARTHDATA site of EOSDIS and its website:
https://disc.gsfc.nasa.gov/datacollection/LIMSN7L3_006.html (Remsberg et al., 2011).  The
SPARC-Data Initiative data are located at https://doi.org/10.5281/zenodo.4265393 (Hegglin et
al., 2021).  We acknowledge the individual instrument teams and respective space agencies for
making their measurements available, and the Data Initiative of WCRP's (World Climate
Research Programme) SPARC (Stratospheric Processes and their Role in Climate) project for
organizing and coordinating the compilation of the chemical trace gas datasets used in this work.

*Author Contributions.*  ER led the manuscript and prepared most of the figures with
contributions from his co-authors.  MH conducted the trajectory study and generated its figures.
EH provided his rocketsonde data on ozone and temperature along with their error estimates.
*Competing interests.*  The authors declare no competing interests for this study.
*Acknowledgements.* The authors appreciate John Gille and Jim Russell III and members of the
LIMS Science Team for their leadership in the development of the LIMS instrument and for their
processing of its historic data products.  The authors are grateful to John Burton, Praful Bhatt,
Larry Gordley, B. Thomas Marshall, and R. E. Thompson for producing the V6 Level 2 dataset.
They acknowledge Gretchen Lingenfelser for her work in generating and archiving the V6 Level
3 coefficient dataset.  They appreciate especially the constructive comments from the two
anonymous referees.  They also thank V. Lynn Harvey for her comments on an early draft of the
manuscript.  EER and MN carried out their work while serving as Distinguished Research
Associates of the Science Directorate at NASA Langley.

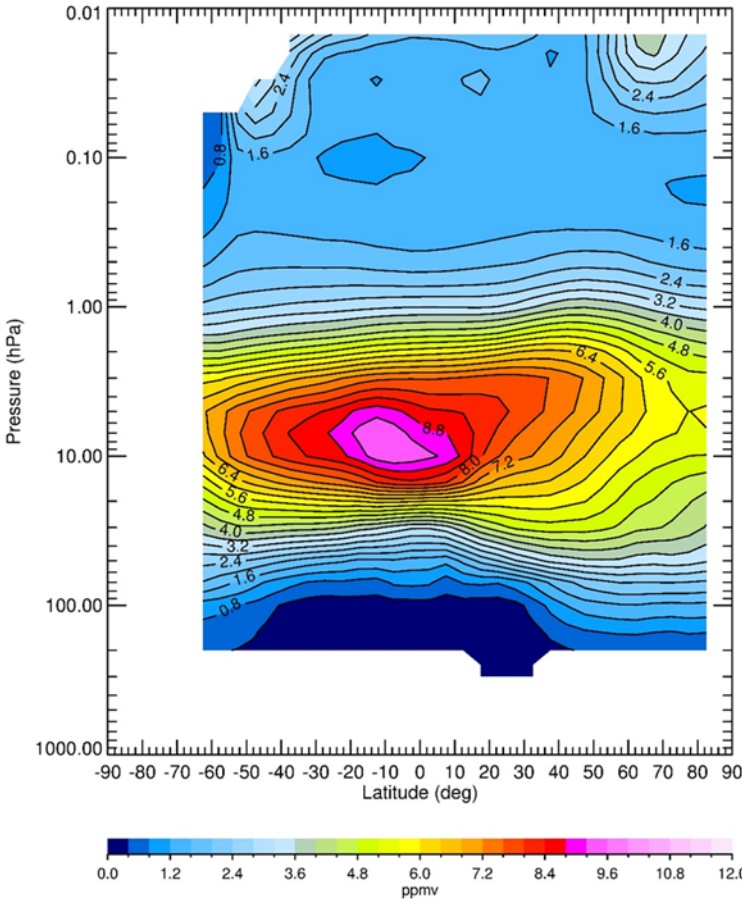


Figure 1—LIMS V6 Level 3 monthly zonal mean ozone for descending-mode only (or nighttime
equatorward of ~55°S) for January 1979.  Contour interval (CI) is 0.4 ppmv.


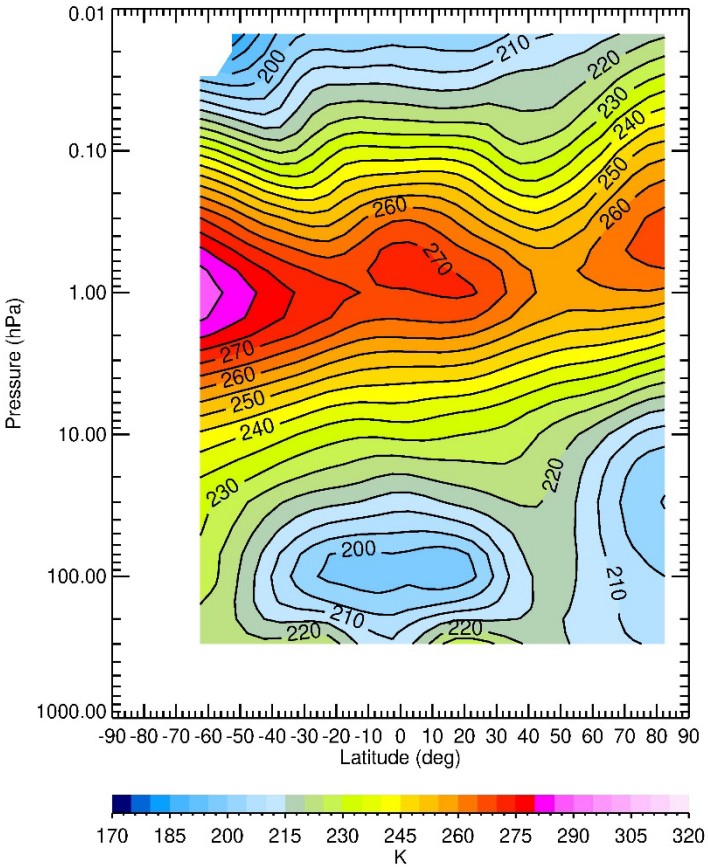


Figure 2—Zonal average, descending-mode, temperature for January 1979.  CI is 5 K.


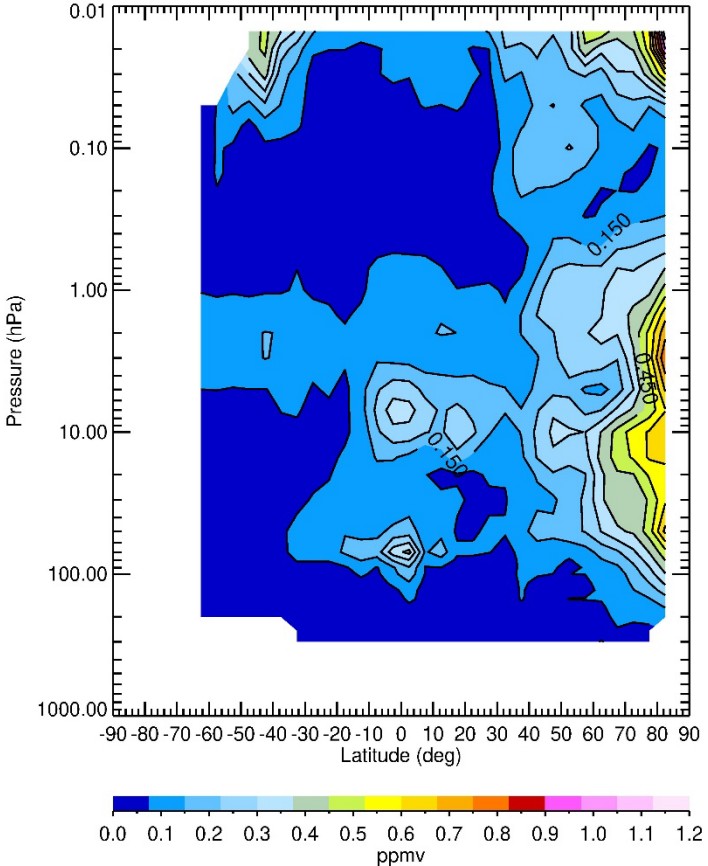



Figure 3—Zonal standard deviation about average (A+D) zonal mean ozone for January 1979.
CI is 0.075 ppmv.


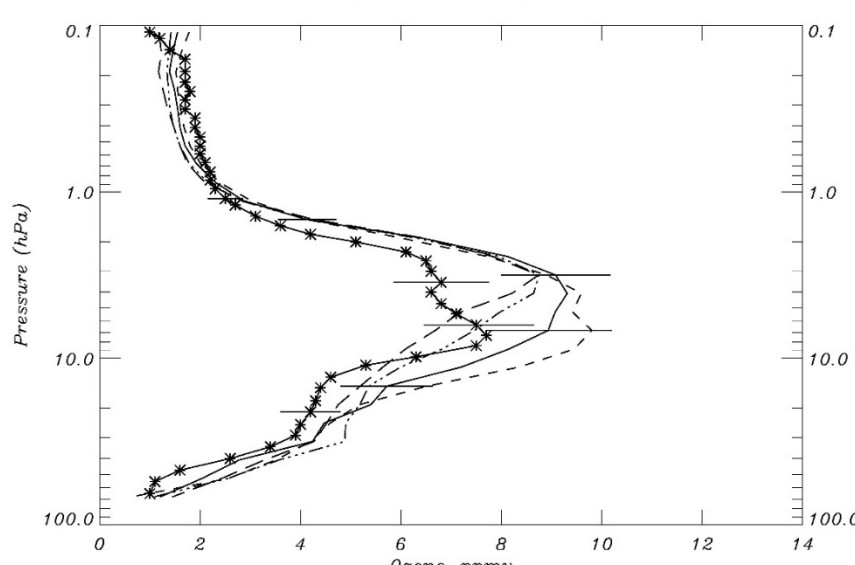

*LIMSv6 O3 (desc) vs chem O3 (*), Dec 15, 1978*


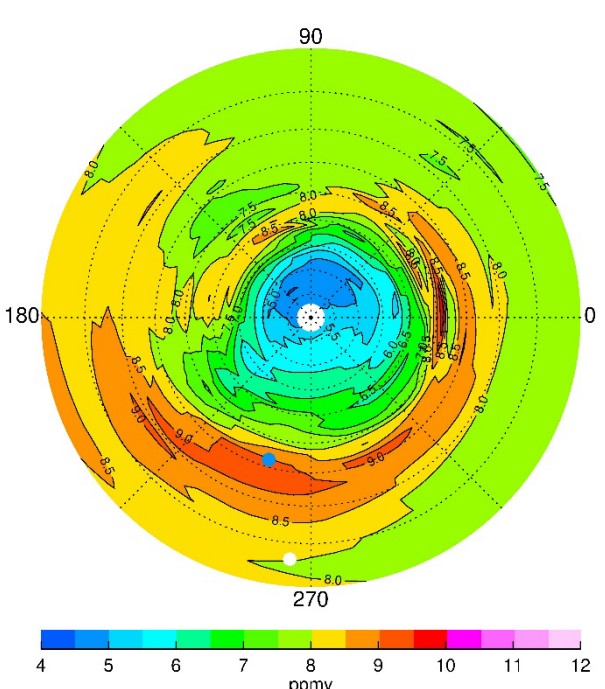


Figure 4—(top) Profiles of V6 ozone (at 0611Z) versus rocket chem ozone (* at 0541Z) on December 15. The four V6 profiles have separations of 2.6° latitude, and the solid curve (at 31.8°N) is closest to White Sands (WS, 32.4°N). Horizontal bars are ozone errors. (bottom) NH V6 ozone at 4.6 hPa; Greenwich (0°E) is at right, and CI is 0.5 ppmv. Latitude (dotted circles) is every 10°. Satellite location is white dot (6°N, 265°E), and WS is blue dot.

539

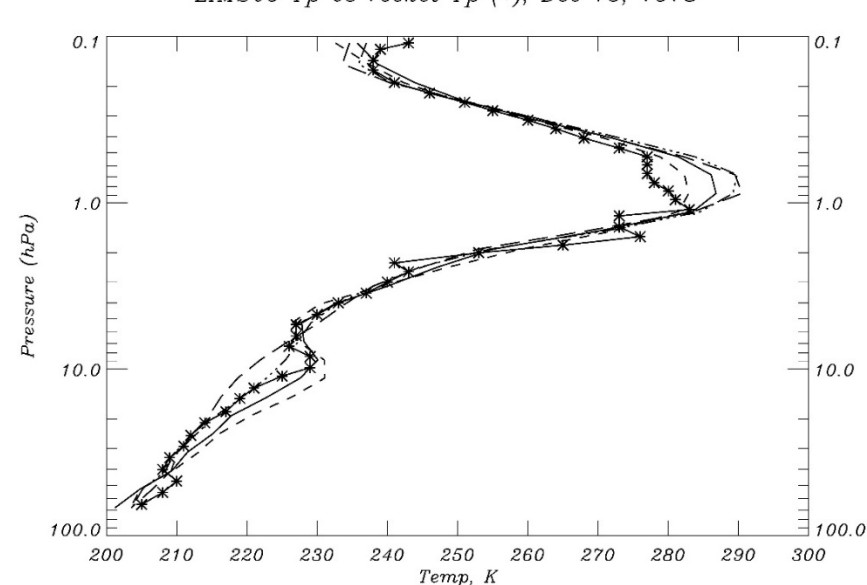

540

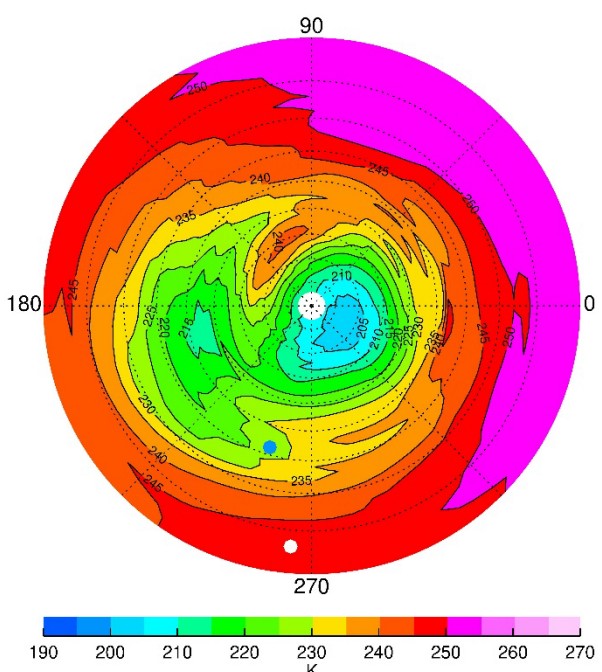

541

Figure 5—(top) Profiles of V6 temperature compared with Datasonde values (*) on December 15. The four V6 profiles are separated as in Fig. 4, where the short-dashed curve is for 29.2° and the long-dashed curve is for 37.2°. (bottom) NH V6 temperature distribution at 4.6 hPa; CI is 5 K, and satellite location is white dot and White Sands is blue dot.


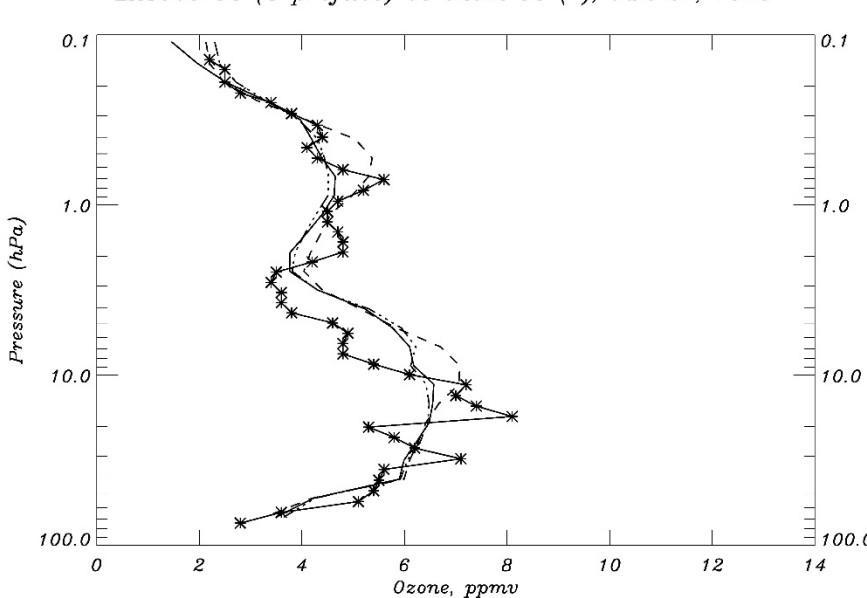

*LIMSv6 O3 (3 profiles) vs chem O3 (∗), Jan 27, 1979*



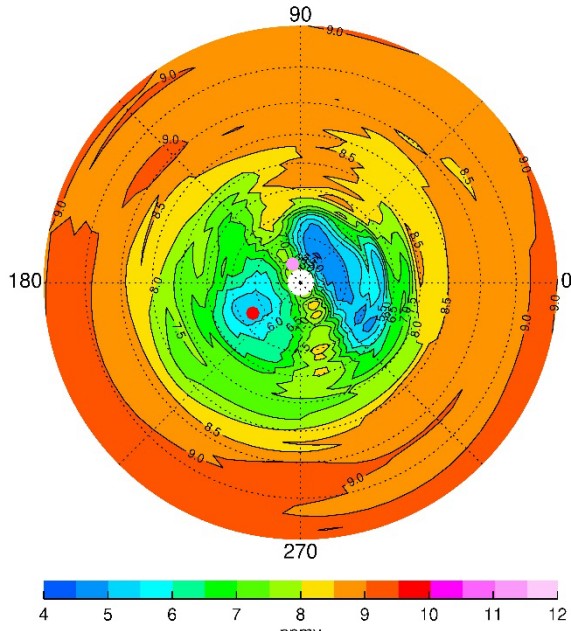


Figure 6—(top) As in Fig. 4, but for January 27, 1979, at Poker Flat, AK (65°N, 212.5°E);

(bottom) NH V6 distribution of ozone at 4.6 hPa, where CI is 0.5 ppmv.  Latitudes (dotted

circles) are spaced every 10°; Poker Flat is red and satellite position (81°N, 113°E) is pink.



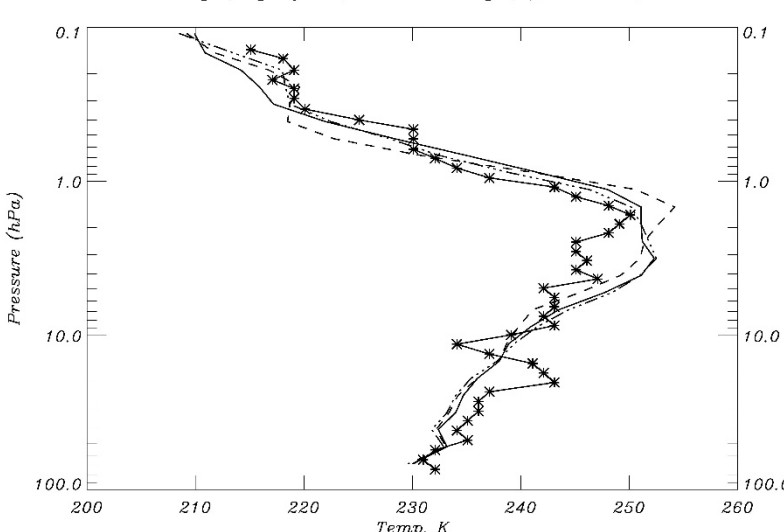

LIMSv6 Tp (3 profiles) vs rocket Tp (*), Jan 27, 1979



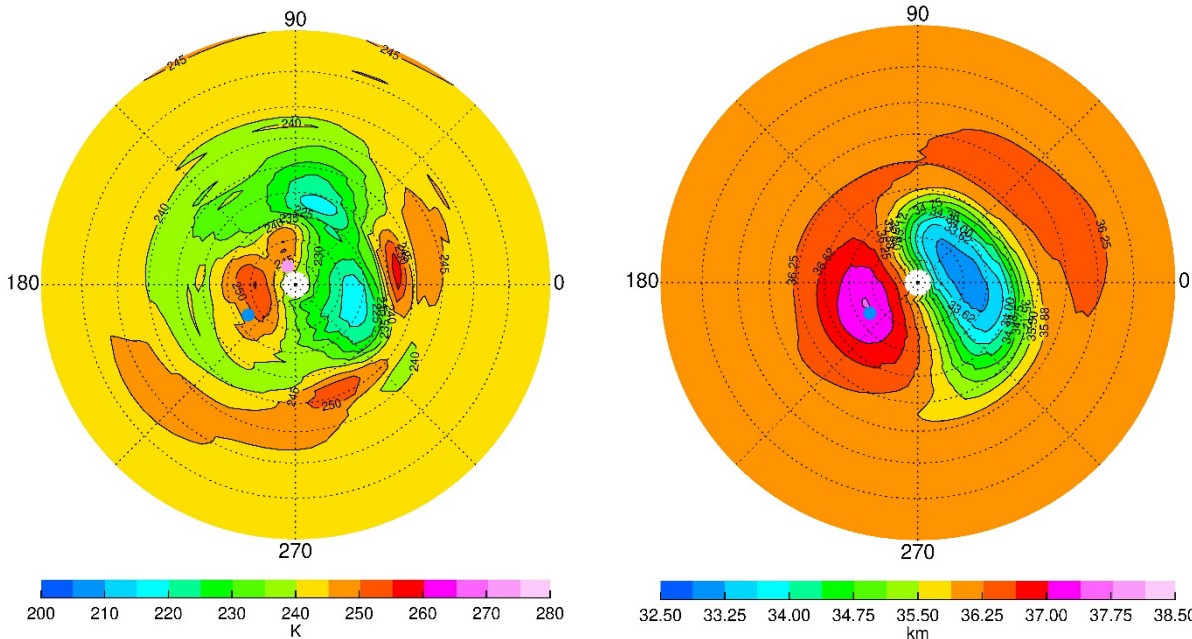



Figure 7—(top) As in Fig. 5, but for January 27, 1979. (bottom-left) NH V6 temperature; CI is 5 K. Poker Flat is blue and satellite position is pink. (bottom-right) V6 GPH; CI is 0.375 gpkm.


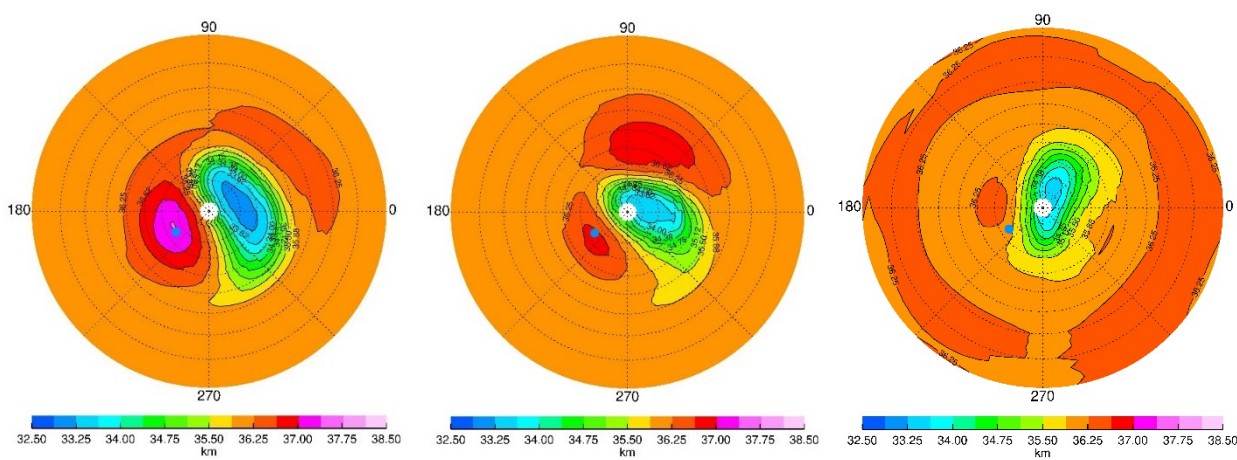


Figure 8—NH V6 GPH at 4.6 hPa; CI is 0.375 gpkm. Poker Flat is blue dot. Panels are spaced
one week apart; (left) February 3; (middle) February 10; and (right) February 17.

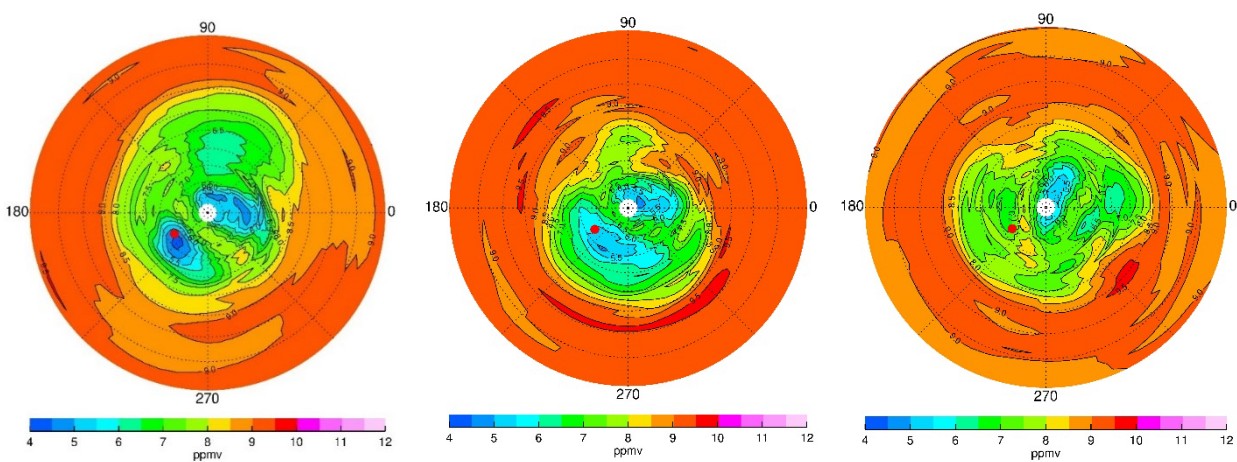



Figure 9—Maps of ozone at 4.6 hPa (left) on February 3, (middle) on February 10; and (right) on
February 17.  CI is 0.5 ppmv and red dot is Poker Flat.

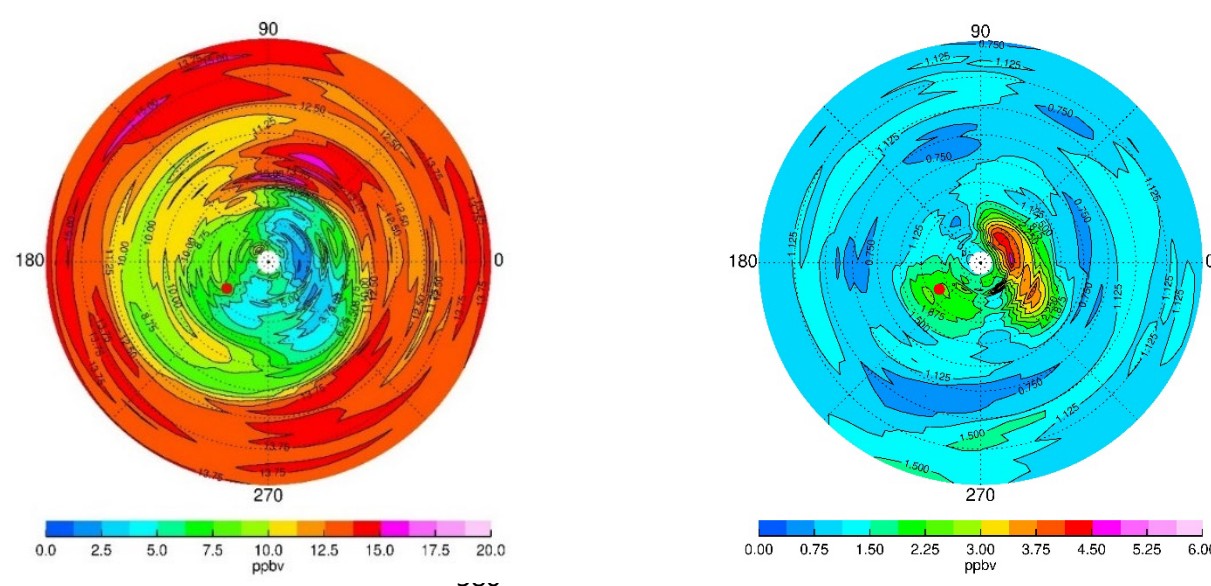



Figure 10—(left) Nighttime NO$_2$ on January 27 at 4.6 hPa; CI is 1.25 ppbv.  (right) HNO$_3$ at 4.6
hPa; CI is 0.375 ppbv.  Red dot is Poker Flat.



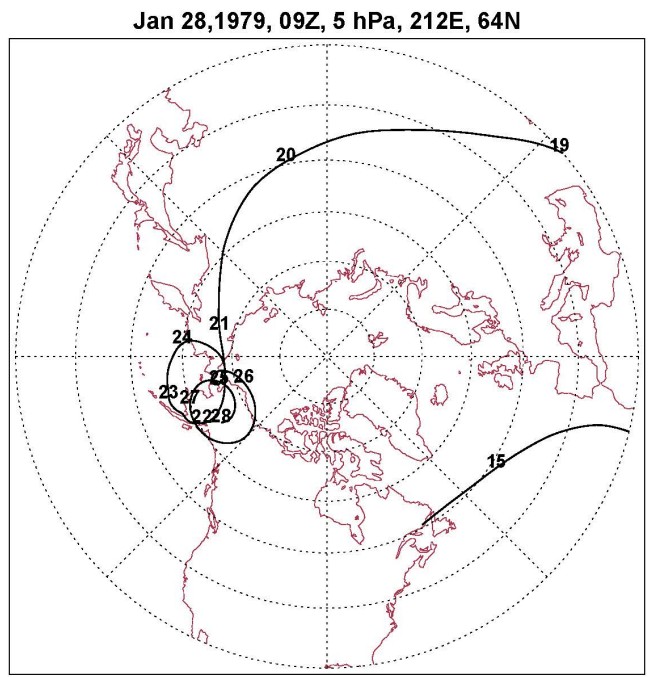

**Jan 28,1979, 09Z, 5 hPa, 212E, 64N**


Figure 11—Trajectory of air parcel that end on January 28 at the location of the LOP.  Numbers
on the trajectory denote the date, beginning with January 15 and where the parcel is equatorward
of 30°N on January 16-18.


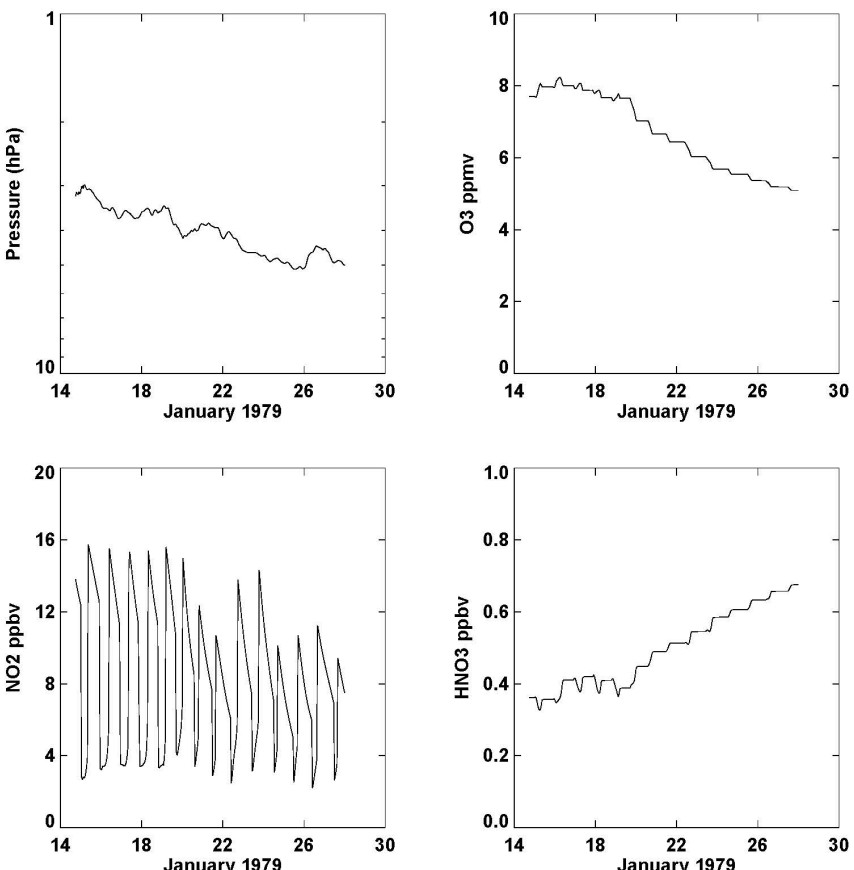


Figure 12—Air parcel history of the changes in its (top left) pressure, (top right) ozone, (bottom
left) $NO_2$, and (bottom right) $HNO_3$.

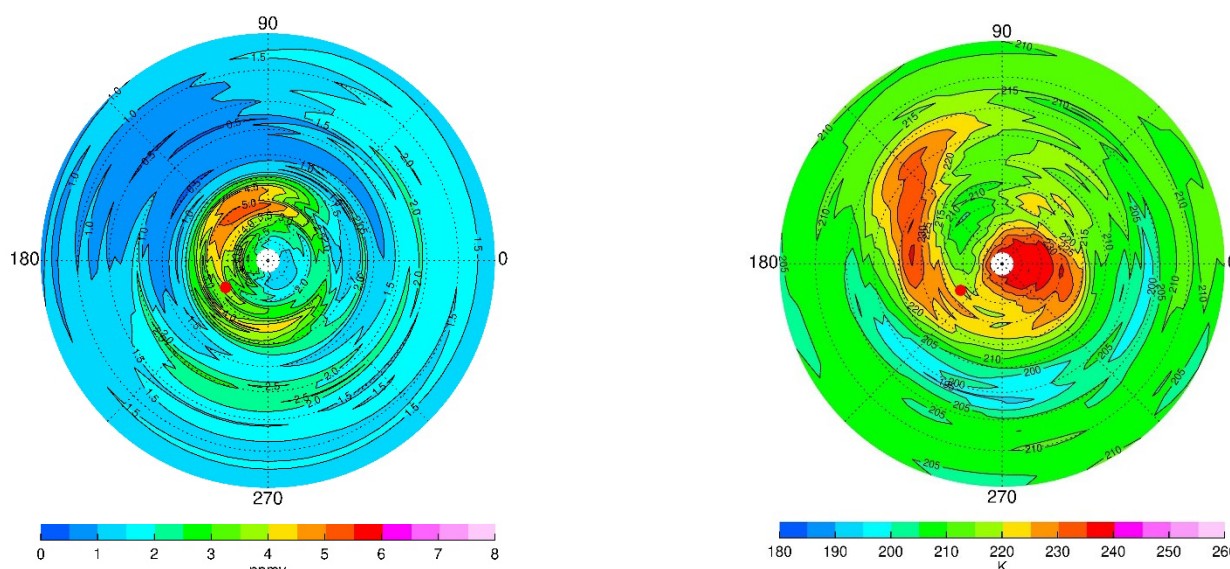


Figure 13—(top) NH distributions for December 15 at 0.022 hPa for (left) ozone and for (right)
temperature; CIs are 0.5 ppmv and 5 K, respectively.  Red dot denotes location of Poker Flat.





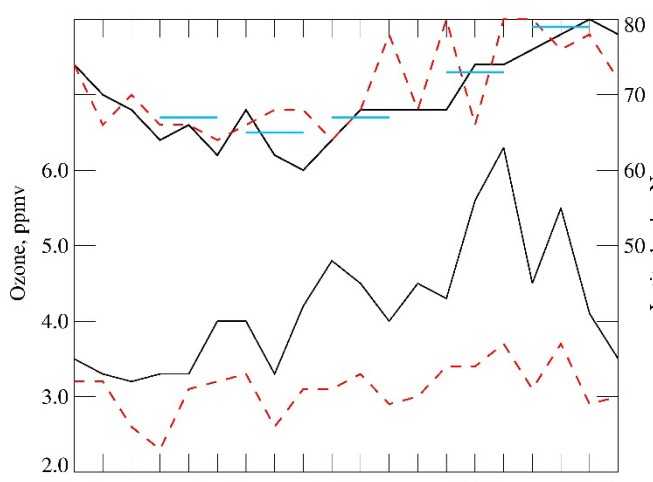


Figure 14—Time series of peak V6 ozone (bottom two curves) and its latitude location (top two
curves) at 0.022 hPa.  Dashed red curves are for combined ozone, while solid curves are for
descending (nighttime) ozone only.  Horizontal blue lines indicate the latitude of the terminator.


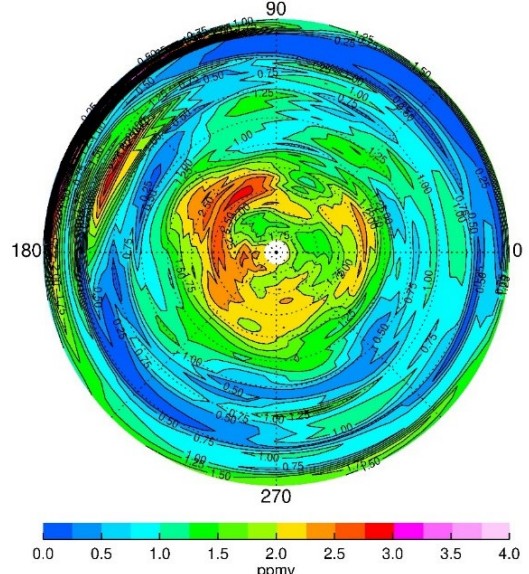



Figure 15—NH V6 combined (A+D) ozone distribution at 68 hPa for December 15, 1978.  CI is
0.25 ppmv.

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
