# Peer review of "Introduction and objectives"

_Atmospheric Measurement Techniques, 2021_

## Referee Comment (RC1)

Referee report for amt-2021-340 (Remsberg et al., Variations of Arctic winter ozone from the LIMS Level 3 dataset)

General Comments:

   I found myself somewhat torn regarding the value of this manuscript, which describes a few features of LIMS Level 3 maps and profiles in the context of the Arctic winter of 1978/1979. The motivation seems to be to generate more visibility for this data set for anyone interested in placing those historical ozone fields (or other fields obtained by LIMS) in "context", given the longer-term changes in and the importance of ozone, in particular.  Most of the usefulness of this nice early data set may well have been "milked", by now, and in large part thanks to the work of the authors of this manuscript. Adding this manuscript at this late stage is not of the highest value, scientifically, or even as a brief data description or as a partial demonstration of validation using Level 3 data. Nevertheless, it is not technically incorrect or flawed, and there may not be enough published research of mesospheric variations, which are reported on to some extent here. I also found that the flow and focus of the manuscript were not that easy to follow. Finally, there are also some data limitations in the case of LIMS (non-LTE effects mentioned in the manuscript) for parts of the upper atmosphere, as mentioned by the authors.
   I do (somewhat marginally) recommend publication in AMT (or a data-type Journal, possibly, if not in AMT), mainly for "historical" reasons. A few minor comments for details and clarity should be addressed (see below); there is nothing major, except for that somewhat "agonizing" part over the worthiness of this publication at this time, since it does not add much to the science and there are clearly more recent studies using many more years of data from other instruments (as referenced in this manuscript), even without the use of synoptic-type maps. It is also not so much of a "measurement technique" type of paper, but this may still be the best option.

Mostly minor/editorial-type comments:

   -   P2, L32, "heights" rather than "height", since this is a sequence of heights.
   -   P3, L53, too many "report on" in these last few sentences of this paragraph. Try using "describe", for example, here, instead.
   -   P3, L56-57, these two sentences use the past tense, and it would be best to use either present or past for the whole paragraph (e.g., use present in these two sentences also).
   -   P3, L61, I think you really just mean "(Level 2)", since there is also a V6 Level 3 data set.
   -   P4, L102, delete "all" in front of "latitudes".
   -   P4, L104, I would delete ", or 33.5 deg counterclockwise…vector" as this is the same sort of statement as the first part of the sentence (but just turned around).
   -   P5, L107, I would use "well registered" rather than "registered well".
   -   P5, L110, it would seem that the latitudinal spacing represented by the samples in Fig. 4 is coarser than 1.6 degrees; is this just the mapping algorithm (coarser) grid [maybe I missed this part]? If this is described well enough in the manuscript, no need to change anything.
   -   P6, L136, no need to redefine SPARC Data Initiative as SPARC-DI (was done earlier), just use one or the other…

- P6, L158, The sentence should be reworded better, e.g. "To first-order, the stratospheric T(p) retrievals account for the effects of horizontal temperature gradients" [+ I would have liked to see a reference regarding the methodology here, even if it might be much more obvious to the authors themselves]. It is hard for the reader to understand this otherwise, and this is either stretching the long-term memories of some or asking too much (literature search) from an interested younger reader.
- P6, L159 (and in general), what do "errors" reflect in this manuscript? Are they estimated 1-sigma-type errors, or double this? Please specify this somewhere (assuming that all error bars represent, say, 1-sigma).
- P6, L161, I suggest a slight rewording: "…bias error for ozone, and these errors grow to about 16% in the middle mesosphere…"
- P7, L173/174: how exactly is it known that the larger SD values are caused by planetary wave activity? Because of their magnitude and extent? Please specify what is known (with a reference, possibly).
- P7, L175/176, this statement would also be better with at least one reference regarding the upward propagation part (and there are certainly references for this).
- P7, L183, "The estimated total error for CHEM…"
- P8, 205: Here, the sonde data are referred to as "Datasonde" rather than chemiluminescent sonde, or just CHEM (as done for Fig. 4 and associated discussions). Either call all the sonde data "CHEM" (short for chemiluminescent sonde), or make it clear when "Datasonde" is introduced that this is the same as "CHEM"... but in my view, one consistent notation (either CHEM or Datasonde) would be better, unless you have a good reason to keep changing notations. I should note that the Hilsenrath (1980) paper never mentions "CHEM", but they do mention chemiluminescent sonde and (one occurrence of) "Datasonde".
- P8, L211, "are well determined along…" [might be better]
- P9, L239, "and relatively low temperatures" [or "and is relatively cold"]
- P9, L244, "at 0.46 hPa or above **in** the Alaskan anticyclone"?
- P10, L257, "from studies of GPH…"
- P10, L258, "They determine the extent…"
- P10, L260, delete the comma before "vertical resolution"
- P11, L263, the vertical resolution has already been defined (3.7 km)
- P11, L286/287, what about downward transport from higher altitudes, is that not also possible / part of the equation?
- P11, L288, NOx includes NO2…so you could delete "(and NO2)"
- P11, L294, "some chemical loss of ozone…"
- P11, L295, "indicates that there were significant variations…"
- P12, L318, "temperatures are much higher in the Canadian sector…"
- P13, L344, delete ", too"
- P14, L371, I would think that with less than one year of data, a baseline is somewhat difficult to establish (given seasonal and QBO effects), but the statement is sort of alright.
- P14, L388/389, it is way too late to reconsider validation efforts for LIMS, in my view, or to add much to past work from such an effort.

- P14, L393/394, this sentence is too nebulous (what does one may find mean?), in large part because this is probably too difficult to accurately assess, given the short period of data from LIMS, in my view. Of course there are changes, but accurately determining an underlying trend requires a good amount of nearly continuous data between "recent decades" and 1979. Also, the community knows that SAGE data have been used for this purpose.
- P15, L396, "surface maps" means what (why not just "maps")?
- Figure 4, one should be able to know which two satellite profiles are immediately adjacent to the CHEM profile. Please specify in the caption.
- P33, L626, please provide all author names for this reference.

---

## Author Comment (AC1)

REPLY TO REFEREE #1 (*in italics*)

Referee report for amt-2021-340 (Remsberg et al., Variations of Arctic winter ozone from the LIMS Level 3 dataset)

General Comments:

   I found myself somewhat torn regarding the value of this manuscript, which describes a few features of LIMS Level 3 maps and profiles in the context of the Arctic winter of 1978/1979. The motivation seems to be to generate more visibility for this data set for anyone interested in placing those historical ozone fields (or other fields obtained by LIMS) in "context", given the longer-term changes in and the importance of ozone, in particular.  Most of the usefulness of this nice early data set may well have been "milked", by now, and in large part thanks to the work of the authors of this manuscript. Adding this manuscript at this late stage is not of the highest value, scientifically, or even as a brief data description or as a partial demonstration of validation using Level 3 data. Nevertheless, it is not technically incorrect or flawed, and there may not be enough published research of mesospheric variations, which are reported on to some extent here. I also found that the flow and focus of the manuscript were not that easy to follow. Finally, there are also some data limitations in the case of LIMS (non-LTE effects mentioned in the manuscript) for parts of the upper atmosphere, as mentioned by the authors.
   I do (somewhat marginally) recommend publication in AMT (or a data-type Journal, possibly, if not in AMT), mainly for "historical" reasons. A few minor comments for details and clarity should be addressed (see below); there is nothing major, except for that somewhat "agonizing" part over the worthiness of this publication at this time, since it does not add much to the science and there are clearly more recent studies using many more years of data from other instruments (as referenced in this manuscript), even without the use of synoptic-type maps. It is also not so much of a "measurement technique" type of paper, but this may still be the best option.

*General comments—We thank the reviewer for a careful assessment of the manuscript, and we understand his/her ambivalence regarding its suitability for AMT.   We initially submitted this manuscript to Earth System Science Data (ESSD) journal, but no associate editor agreed to handle it.  Therefore, we opted to send it to AMT.  To improve the flow of the manuscript, we are moving several figures to Supplemental materials—Fig. 8 that showed lower mesospheric ozone and temperature on January 27 (now Fig. S1) and figures showing three panels of H2O and three panels of temperature at 0.022 hPa (Figs. S2 and S3).  A separate important aspect of the V6 Level 3 dataset is that its daily maps show more clearly the strong horizontal gradients at the polar and subtropical edges of  ozone streamers, at least compared with those presented in Leovy et al. (1985)--see example in Fig. S4.  We rescaled the panels in Figs. 4-7 and rearranged the panels in Figs. 9-11.  New Fig. 9 now shows two NO2 panels—one at 4.6 hPa and another at 3.2 hPa, indicating that there may have been some downward transport of NO2 in the region of the LOP. Fig. 9 also shows a relative maximum in $HNO_3$ at 4.6 hPa.  Figure 10 shows the three ozone panels that were in original Fig. 10.  Figure 11 has an ozone and a temperature panel—for December 15, showing the relation of temperature with the tertiary ozone feature but now based on combined (A+D) ozone.  Three more ozone and their corresponding temperature panels are in Figs. S2 and S3 to indicate their changing structure across the Arctic region during winter.  Fig. 12 is new and*

*shows a time series of the tertiary ozone feature. Figures 9-12 provide more insight about the value of the V6 dataset for science studies of the separate LOP and tertiary ozone features.*

*The relevant new figures are at the end of our response.*

Mostly minor/editorial-type comments:

*We have incorporated your editorial suggestions and/or added a reference, where needed. In addition, we comment on several of your specific concerns/questions in the following.*

- P2, L32, "heights" rather than "height", since this is a sequence of heights.
- P3, L53, too many "report on" in these last few sentences of this paragraph. Try using "describe", for example, here, instead.
- P3, L56-57, these two sentences use the past tense, and it would be best to use either present or past for the whole paragraph (e.g., use present in these two sentences also).
- P3, L61, I think you really just mean "(Level 2)", since there is also a V6 Level 3 data set.
- P4, L102, delete "all" in front of "latitudes".
- P4, L104, I would delete ", or 33.5 deg counterclockwise…vector" as this is the same sort of statement as the first part of the sentence (but just turned around).
- P5, L107, I would use "well registered" rather than "registered well".
- P5, L110, it would seem that the latitudinal spacing represented by the samples in Fig. 4 is coarser than 1.6 degrees; is this just the mapping algorithm (coarser) grid [maybe I missed this part]? If this is described well enough in the manuscript, no need to change anything.

*P5, Line 110--You raise a good point. Each LIMS up/down, horizon scan pair yields a single retrieved V6 profile that is separated from the next profile by 144 km along the orbital tangent track (or by 1.3° at low to middle latitudes, instead of 1.6°). The V6 data in Fig. 4 (top) are for every other profile along the viewing track near White Sands and have spacings of 2.6°. The Level 3 zonal coefficients were analyzed at every 2° of latitude, based on tangent track profiles closest to that latitude. We have revised the manuscript, accordingly.*

- P6, L136, no need to redefine SPARC Data Initiative as SPARC-DI (was done earlier), just use one or the other…

- P6, L158, The sentence should be reworded better, e.g. "To first-order, the stratospheric T(p) retrievals account for the effects of horizontal temperature gradients" [+ I would have liked to see a reference regarding the methodology here, even if it might be much more obvious to the authors themselves]. It is hard for the reader to understand this otherwise, and this is either stretching the long-term memories of some or asking too much (literature search) from an interested younger reader.
- P6, L159 (and in general), what do "errors" reflect in this manuscript? Are they estimated 1-sigma-type errors, or double this? Please specify this somewhere (assuming that all error bars represent, say, 1-sigma).

*P6, Line 159--V6 profile errors are root-sum-squared (RSS) errors, as defined from the LIMS error analysis studies. We added the reference, Remsberg et al. (2021).*

- P6, L161, I suggest a slight rewording: "…bias error for ozone, and these errors grow to about 16% in the middle mesosphere…"
- P7, L173/174: how exactly is it known that the larger SD values are caused by planetary wave activity? Because of their magnitude and extent? Please specify what is known (with a reference, possibly).

*P7, Lines 173/174--Monthly SD values are the zonal standard deviations about daily zonal means, followed by taking their monthly average. Although gravity waves are not of the scale of planetary waves, they are of a spatial scale at Arctic latitudes that contributes to zonal SD values.*

- P7, L175/176, this statement would also be better with at least one reference regarding the upward propagation part (and there are certainly references for this).
- P7, L183, "The estimated total error for CHEM…"
- P8, 205: Here, the sonde data are referred to as "Datasonde" rather than chemiluminescent sonde, or just CHEM (as done for Fig. 4 and associated discussions). Either call all the sonde data "CHEM" (short for chemiluminescent sonde), or make it clear when "Datasonde" is introduced that this is the same as "CHEM"... but in my view, one consistent notation (either CHEM or Datasonde) would be better, unless you have a good reason to keep changing notations. I should note that the Hilsenrath (1980) paper never mentions "CHEM", but they do mention chemiluminescent sonde and (one occurrence of) "Datasonde".

*P8, Line 205--We no longer make use of the acronym CHEM. The Datasonde provides the temperature profile from a nearly co-located, separate rocket sounding.*

- P8, L211, "are well determined along…" [might be better]
- P9, L239, "and relatively low temperatures" [or "and is relatively cold"]
- P9, L244, "at 0.46 hPa or above **in** the Alaskan anticyclone"?
- P10, L257, "from studies of GPH…"
- P10, L258, "They determine the extent…"
- P10, L260, delete the comma before "vertical resolution"
- P11, L263, the vertical resolution has already been defined (3.7 km)

- P11, L286/287, what about downward transport from higher altitudes, is that not also possible / part of the equation?

*P11, Lines 286/287-- Downward transport may be happening in the region of the LOP. To infer that, we now show two maps of NO2 in revised Fig. 9—one at 4.6 hPa (as before) and another at 3.2 hPa—and a map of HNO3 at 4.6 hPa.*

- P11, L288, NOx includes NO2…so you could delete "(and NO2)"
- P11, L294, "some chemical loss of ozone…"
- P11, L295, "indicates that there were significant variations…"
- P12, L318, "temperatures are much higher in the Canadian sector…"
- P13, L344, delete ", too"
- P14, L371, I would think that with less than one year of data, a baseline is somewhat difficult to establish (given seasonal and QBO effects), but the statement is sort of alright.
- P14, L388/389, it is way too late to reconsider validation efforts for LIMS, in my view, or to add much to past work from such an effort. P14, L393/394, this sentence is too nebulous (what does one may find mean?), in large part because this is probably too difficult to accurately assess, given the short period of data from LIMS, in my view. Of course there are changes, but accurately determining an underlying trend requires a good amount of nearly continuous data between "recent decades" and 1979. Also, the community knows that SAGE data have been used for this purpose.

*P14, L393/394--We delete this sentence and no longer emphasize the use of V6 data for long-term trend studies.*

- P15, L396, "surface maps" means what (why not just "maps")?
- Figure 4, one should be able to know which two satellite profiles are immediately adjacent to the CHEM profile. Please specify in the caption.
- P33, L626, please provide all author names for this reference.

*Figures--*

*Below we show new Figure 12.  We also show the second NO2 panel (for 3.2 hPa) and an HNO3 panel at 4.6 hPa for revised Figure 9.  Fig. S4 of the Supplemental Materials compares V6 ozone for January 27 at 10 hPa with a similar map from Leovy et al. (1985).*

[Figure]

*New Figure 12--Time series of peak V6 daily ozone at 0.022 hPa and its latitude location, as plotted every 7 days.*

*The time series are for peak ozone (bottom two series) and their latitude locations (top two).  Dashed red curves represent zonal mean results for the combined (A+D) data; solid black curves are results for nighttime (D) only.  Blue horizontal lines represent latitude position of the terminator at 30 km altitude.*

[Figure]

*Figure 9—Additional panels (at left) of NO2 at 3.2 hPa and (at right) of HNO3 at 4.6 hPa on January 27.*

*New figure for Supplemental Materials--*

[Figure]

*Fig. S4—Comparison of ozone at 10 hPa for 27 January from (left) V6 versus (right) Leovy et al. (1985).*

---

## Author Comment (AC2)

Reply to Referee #2 (*in italics*)

Report for manuscript AMT-2021-340 on "Variations of Arctic winter ozone from the LIMS Level 3 dataset" by Remsberg et al.

General comments:

I understand that the major focus of the paper is to demonstrate "the value and use" of the LIMS V6 Level 3 data of the arctic winter 1978-1979. In doing so the authors try to show that some O3 phenomena and characteristics, found in posterior analysis of more recent (and some more complete) datasets, are also present in the LIMS V6 L3 dataset (a clear example of this is Sec. 5).

From the point of view of science, I see no aspect which is really new. On the other hand, to show that some O3 features are also present in LIMS data is useful, as this is an independent dataset. Hence, although I cannot see any major scientific contribution I cannot see any either strong reason for not publishing it -the manuscript is very well written-. It is a shame that some of these phenomena have not been published before using LIMS data.

One possibility to enhance the manuscript value would be to compare more quantitatively the variations/characteristics found in LIMS with previous studies. This will be more useful for readers, instead of just showing "... some LIMS examples of the larger-scale variations of Arctic ozone, temperature, and GPH".

On another note, I am not fully convinced that this paper falls completely in the AMT scope. The main aim of the manuscript is not to present the LIMS L3 dataset, which it seems has been published before (Remsberg et al., AMT, 2021; Remsberg et al., 2011; Remsberg and Lingenfelser, 2010), but some O3 phenomenology.

*General comments: Thank you for your thorough review and for adding some references. We originally submitted our manuscript to Earth System Science Data (ESSD) journal, but it did not attract an associate editor after a wait of more two months. We believe that AMT is an appropriate alternate venue. In the revised text, original Fig. 8 is moved to Supplemental materials as Fig. S1 to achieve better continuity of the subject matter in the manuscript. The maps of GPH are now Figure 8. We relate the late January/early February LOP feature to that in Morris et al. In Figure 9 we now show two maps of NO2— at 4.6 hPa and at 3.2 hPa—to indicate that there may be downward transport of NO2 from 3.2 hPa to 4.6 hPa in the region of the LOP. Fig. 9 also shows a relative increase in HNO3 at 4.6 hPa in that region. The three ozone maps in original Fig. 10 are retained in new Fig. 10.*

*LIMS provides high northern latitude measurements throughout winter, unlike SABER.  The stratopause from LIMS V6 zonal mean temperatures is near 55 km at 70-84N for November and December 1978 but then shifts back to just below 50 km for January and February 1979.  In that respect, the LIMS winter of 1978-79 appears normal, compared with the several anomalous winters reported in Smith et al. (2009). Our study of the tertiary ozone maximum is possible with V6 because its ozone profiles extend to the upper mesosphere.  Zonal mean ozone has a tertiary maximum at about 0.022 hPa (~73 km).  Fig. 11 shows maps that indicate the zonal variability of temperature and the tertiary ozone feature at 0.022hPa for December 15.  New Figure 12 is a time series of the peak ozone and its latitude location for one day of each week from November through mid-March.  Peak zonal mean ozone occurs in early February, which differs somewhat from that of other datasets.  Maps of ozone and temperature for three other days that winter are in the Supplementary material (Figs. S2 and S3); the structure and continuity of the temperature features appear related to an advection process, as opposed to uncorrected NLTE effects.*

*We also include Fig. S4 (in Supplementary materials), comparing the map of V6 ozone for January 27 at 10 hPa, based on the mapping of the V6 profiles, versus that shown in Leovy et al. (1985) from an earlier LIMS map version.  V6 displays tighter ozone gradients along its subtropical boundary, primarily because the SE mapping algorithm for V6 has a short memory (or relaxation time) of before and after January 27.*

Minor/moderate comments:

*We agree with your editorial suggestions and have made corrections or added a reference, where needed.  We comment on several of your specific concerns/questions below.*

P2, L27-28, I do not understand this sentence. V6 are satellite measurements. Hence, I do not understand why "V6 satellite data" "are important for interpreting satellite limb infrared measurements versus local measurements." Maybe the authors want to say that LIMS V6 are important for interpreting other (non-satellite) "local" measurements?

*We altered the sentence to read—"We illustrate how the synoptic maps of V6 ozone and temperature are an important aid…"*

P2, L41, For many readers the middle atmosphere includes the mesosphere. This sentence should be re-written. Something like: "Ozone is an excellent tracer of the stratosphere (or lower stratosphere)".

P3, L52, I suggest adding also the SABER observations (Smith et al. GRL, 2009).

P5, L115-116, LIMS V6 free of non-LTE below ~0.05 hPa. This is true for most conditions except in the polar winter regions, (or during strat-warm) where it is expected to be significant (see, Fig. 22d in Funke et al., 2012).

*We now cite the NLTE study of Funke et al. (2012) here and at Line 141. Thank you for pointing it out.*

P6, L141-142, see comment above. The data might be affected by NLTE even at night.

P6, L147-148, "A tertiary ozone maximum is present in the upper mesosphere near the day/night terminator zones of the LIMS measurements for January (~50°S …". This seems very interesting. However, such a tertiary maximum is not present in MIPAS measurements in January in the Southern hemisphere (e.g. ~50ºS) (see Fig. 12 in Lopez-Puertas et al., 2018). It is not present either in the Southern hemisphere winter, e.g. July near 50ºN. Also, I have not seen this kind of enhancement in otehr O3 datasets. Those conditions are polar summer. Should we expect a tertiary maximum in summer conditions? Could the authors check this behaviour. If it is found to be real it would be very useful to comment in the manuscript about the reasons for the maximum in those regions.

*The inclusion of 50S in line 148 was a misstatement, and we are deleting it. The rapid change with latitude near 50S is because LIMS was viewing across the night/day terminator in January, as you note in your comment about P6, L154.*

P6, L150, "The location (~0.02 hPa) and magnitude (~3.5 ppmv) of the NH maximum agree with those reported from subsequent satellite studies by Smith et al." I would probably say slightly larger: MIPAS values are always below ~2.5 ppmv (Smith et al., 2018, Fig. 4 and Lopez-Puertas et al., 2018, Fig. 12).

*Peak, zonal mean (A+D) ozone values in new Fig. 12 range from 2.2 ppmv on November 8 at 66°N to 3.7 ppmv on February 14 at 80°N and on February 28 at 76°N. Peak V6 nighttime ozone values are larger but also more variable. Those LIMS values are larger than ones from MIPAS and from AURA-MLS.*

P6, L154, "Thus, the decrease of mesospheric V6 ozone at 0.1 hPa and poleward of 60°S in Fig. 1 indicates merely a change from night to day values''. That is correct. The diurnal variation of O3 is clearly seen, for example, in Figs. 11, 12 and 13 of Lopez-Puertas et al., 2018.

P8, L199-200. Could the authors comment on the differences in local time between the rocket O3 measurements and the satellite measurements? They could lead to significant differences in O3 (see, e.g. Studer et al., 2014, Figs. 4a).

*Observed diurnal differences in ozone are smaller than the differences between V6 and rocket at White Sands in Fig. 4. We now include the time difference (~half an hour) along with the latitude coordinates of the V6 and rocket soundings. Most likely the measurements are not exactly co-located (limb path versus a local sounding) and/or there were inadequate gradient corrections for the V6 profiles.*

P8, L205. Which is the meaning of the asterisk?

*The asterisk refers to the Datasonde profile in caption of Fig.5.*

P8. L211-212. I do not understand this point. Temperature differences between datasonde and V6 at ~0.5-1 hPa are significant, close to 10K, but O3 compares well. How can this be explained?

*You raise a good point. The map shows that there is a strong gradient in temperature above White Sands, and separate maps (not shown) from the descending versus the ascending orbital scans indicate differences of ~5K at that spot, an indication of not having corrections for T(p) gradients for V6 in the mesosphere. Yet, ozone compares well at 0.68 hPa, as you say. One possible explanation is that the correlative ozone and temperature measurements are from separate instruments on different rocket soundings—there may be a co-location issue between them. Results of this kind indicate how difficult it is to obtain good correlative comparisons during disturbed atmospheric situations.*

P9, L232-233. I do not fully understand the aim of this sentence. Is it that "V6 ozone has very little bias due to temperature" (the temperature measured by LIMS I guess)? I believe this has been verified before, in validation studies. Otherwise, I think the authors should not reach this important conclusion from just comparing a few profiles which, btw, differ by more than 5 K.

*The sentence will be revised. According to the bias estimates in Remsberg et al. (2021, their Table 1), retrieved V6 ozone in Fig. 6 should be affected significantly by the temperature differences in Fig. 7 (i.e., a +5K bias would impart a nearly -40% ozone error at 3 hPa). The observed smaller ozone differences between V6 and sonde may again be due to co-location differences between the separate rocket ozone and temperature soundings in this disturbed atmospheric region and/or to the spatial differences of the V6 tangent layer versus in situ measurements.*

P10, L263-264. Are the authors suggesting that LIMS data would be useful to study LOPs in the mesosphere? I think it is not the case. O3 should not be considered a good tracer in the mesosphere.

*Your concern is valid, and we no longer describe ozone as a tracer for the LOP in the lower mesosphere.*

P11, L287-288. It seems to me rather descriptive and a bit speculative. To confirm this would require a quantitative analysis. Further, this contrasts with the idea mentioned above that O3 can be considered as a good tracer in the mid-stratospheric arctic region.

*Revised Fig. 9 now contains two plots of nighttime NO2 only, one at 4.6 hPa, as before, and another at 3.2 hPa showing equivalent values of NO2 in the small region of the LOP and indicating possible descent and consequent loss of ozone due to chemical processes. We added a plot of $HNO_3$ to Fig. 9, and it is also elevated in the region of the LOP at 4.6 hPa. This is an indication of chemical conversion of $NO_2$ to $HNO_3$ at the center of the anticyclone.*

P12, L312. It would be useful to draw the terminator in the upper panels of Fig. 11.

*We now include the latitude of Poker Flat (red dot at 65°N) as a reference location, and we show the position of the terminator in new Fig. 12 (see following response also).*

P12, L312-313. Can the behaviour shown, derived from two single days in different months, be considered as representative of the tendency along the winter? E.g. an increase of the O3 tertiary maximum as the winter progresses? MIPAS O3 shows no clear tendency and it varies from year to year (see Fig. 15, bottom/right panel of Lopez-Puertas et al., 2018). Also, the data reported by Smith et al. (2018) shows that the O3 tertiary maximum decreases in Feb (see their Fig. 3, right panels).

*New Figure 12 is a time series of peak ozone and its latitude location at 0.022 hPa. The position of the terminator is noted, as well, and it shifts toward higher latitudes away from winter solstice. Peak zonal mean values increase slowly from a minimum of 2.2 ppmv in November, to ~3.1 ppmv in January, to a maximum of 3.7 ppmv in February, and then declining to 3.0 ppmv by mid-March. The enhanced values in February follow the minor SSW of late January and the final SSW event of mid to late February.*

P12, L329-330. About the sentence "Although the seasonal evolution of the tertiary ozone maximum is understood reasonably well (Smith et al., 2018), there is more information about this ozone feature from the daily maps of ozone, T(p), and GPH from Level 3.", Could the authors clarify which "more information" is in LIMS data which is not available from later sensors (e.g. SABER, MIPAS, GOMOS, ACE, etc.) that also measures O3 globally, over longer time scales, with more extended altitude ranges and with better sensitivity (see, e.g., Smith et al. 2013). Many of those instruments also measured daily maps. I understand "more information" in the sense that it provides very important measurements taken more than two decades before, in the winter of 1978-1979, but not in the other respects.

*We modified the sentence as "one can gain more information…from the daily maps…". Certainly, there is also more information from the more recent satellite datasets.*

Some suggestions for the figures and figures captions:

Fig. 1. What are the conditions for lat. >55ºS? Only daytime?

*Yes, in daylight.*

Fig. 3. Zonal "mean"? Maybe the caption could be made more explicitly, as in the text.

*Should be "zonal standard deviations about the average (A+D) zonal mean…"*

Fig. 4. I suggest specifying the illuminations conditions (day? night? local solar time of the CHEM sonde?). Also, please add the text included in the caption of Fig. 5 ( "where the short-dashed curve is for 29.2° and the long-dashed curve is for 37.2°").

Fig. 5. Why use a different pressure level for the temperature map than for O3 in Fig. 4?

*We want to show what the temperature field looked like, where there is a discrepancy in T(p) between the V6 and Datasonde profiles.*

Fig. 6. Please add the sentence about the altitudes as in Fig. 4: ("Latitudes (dotted circles) are spaced every 10°.").

Fig. 7, top panel. Improve contrast, make axis lines and marks thicker.

*The .jpeg figure is clear, but it was degraded in the .pdf manuscript.  We will check that it looks OK for the published manuscript.*

Fig. 12. Is O3 for daytime? nighttime? both?

*Original Fig. 12 is now Fig. 13.  Its ozone is from the combined (or A+D) orbital profile data.*

References

Funke, B., López-Puertas, M., Garcia-Comas, M., Kaufmann, M., Höpfner, M., and Stiller, G. P.: GRANADA: a generic RAdiative traNsfer AnD non-LTE population algorithm, J. Quant. Spectros. Radiat. Transfer, 113, 1771–1817, https://doi.org/doi: 10.1016/j.jqsrt.2012.05.001, 2012.

López-Puertas, M., García-Comas, M., Funke, B., Gardini, A., Stiller, G. P., Clarmann, T. von, Glatthor, N., Laeng, A., Kaufmann, M., Sofieva, V. F., Froidevaux, L., Walker, K. A., and Shiotani, M.: MIPAS observations of ozone in the middle atmosphere, Atmos. Meas. Tech., 11, 2187–2212, https://doi.org/10.5194/amt-11-2187-2018, 2018.

Smith, A. K., López-Puertas, M., García-Comas, M., and Tukiainen, S.: SABER observations of mesospheric ozone during NH late winter 2002–2009, 36, L23804, https://doi.org/10.1029/2009GL040942, 2009.

Smith, A. K., Harvey, V. L., Mlynczak, M. G., Funke, B., Garcia-Comas, M., Hervig, M., Kaufmann, M., Kyrölä, E., López-Puertas, M., McDade, I., Randall, C. E., Russell III, J. M., Sheese, P. E., Shiotani, M., Skinner, W. R., Suzuki, M., and Walker, K. A.: Satellite observations of ozone in the upper mesosphere, Journal of Geophysical Research, 118, 5803–5821, https://doi.org/10.1002/jgrd.50445, 2013.

Studer, S., Hocke, K., Schanz, A., Schmidt, H., and Kämpfer, N.: A climatology of the diurnal variations in stratospheric and mesospheric ozone over Bern, Switzerland, Atmos. Chem. Phys., 14, 5905–5919, https://doi.org/10.5194/acp-14-5905-2014, 2014.

*We now cite the first three references, but not last two.  Smith et al. (2013) does not characterize the tertiary ozone feature.  Studer et al. (2014) report on diurnal variations in ozone, but the V6 and rocket ozone profiles in Fig. 4 are separated by only half an hour.*

*Figures--*

*Below we show new Figure 12. We also show the second NO2 panel (for 3.2 hPa) and an HNO3 panel at 4.6 hPa for revised Figure 9. Fig. S4 of the Supplemental Materials compares V6 ozone for January 27 at 10 hPa with a similar map from Leovy et al. (1985).*

[Figure]

*New Figure 12--Time series of peak V6 daily ozone at 0.022 hPa and its latitude location, plotted at every 7 days.*

*The time series are for peak ozone (bottom two) and their latitude locations (top two). Dashed red curves represent zonal mean results for the combined (A+D) data; solid black curves are results for nighttime (D) only. Blue horizontal lines represent latitude position of the terminator at 30 km altitude.*

[Figure]

*Figure 9—Additional panels (at left) of NO2 at 3.2 hPa and (at right) of HNO3 at 4.6 hPa on January 27.*

*New figure for Supplemental Materials--*

[Figure]

*Fig. S4—Comparison of ozone at 10 hPa for 27 January from (left) V6 versus (right) Leovy et al. (1985).*

---

## Author Response (AR2)

Author Reply (*in italics*) to AMT Report #1 of January 29

**Suggestions for revision or reasons for rejection (will be published if the paper is accepted for final publication)**

I am glad to hear that my comments have been useful to the authors and that they have considered them all. I am satisfied with the new version and hence I recommend its publication in AMT.
There are a couple of minor points that the authors may still want to consider.

1) Fig.1. The authors clearly explain (and revise the text, L151-153) the low values found at high altitudes at latitudes polewards of 50S.
However, Fig. 1 also shows a maximum at 30S-50S of similar magnitude and at very similar pressure levels than the tertiary maximum discussed in the NH near ~67ºN. I think this region is not affected by the night/day terminator. Since this enhancement is rather unusual in comparison with measurements of other instruments I think they authors should comment on it.

2) Lines 370-372. About the new Fig. 12 and the evolution of the tertiary maximum in this winter. I think the high value of ~6 ppmv of LIMS is realistic and its difference (larger) compared to those reported in Lopez-Puertas et al. (2018) can be reasonably well explained. First, it is not clear to which plot of that reference the authors compared to. One should have in mind most of the data plotted in that work are monthly means and in some cases averaged over several years. The other reason is given by the authors in their reply:
"The enhanced values in February follow the minor SSW of late January and the final SSW event of mid to late February." MIPAS data for 11 Feb 2009, when the stratopause was rather elevated after the large strat-warm of that winter, show O3 peak values of 5.7 ppmv. Note also that SABER data (Smith et al. 2009, Fig. 2, bottom panel) show enhanced O3 tertiary values at 70-83N in 2009 after the strat-warm. Hence, I think LIMS data in Fig. 12 are very reasonable and in good general agreement with other measurements. The strat-warms induce significant enhancements.

*1—Figure 1 and lines 151 ff—Thank you for raising this concern.  We agree that there is an ozone anomaly in the upper mesosphere near 45S.  We now include a sentence about that feature here, in more detail in the last paragraph of Section 6, and with the aid of Figure S5 (see below) of the Supplemental Materials.  We believe that the enhanced ozone is an artifact of not accounting for path gradients in the retrieval of V6 temperature and from the use of the incorrect temperatures for the retrieval of ozone.*

*2—lines 370-372 and Fig. 12 (which is now Fig. 14)—We considered Lopez-Puertas et al. (2018, their Fig. 15), but we have not looked at daily MIPAS or SABER ozone data, as you may have. Accordingly, we made minor changes to the last paragraph of Section 5 to be clear about that.*

*Additional changes—We removed statements of uncertainty in Section 4 about the chemical changes for air parcels that end up in the region of the LOP. After we submitted our revised manuscript, we conducted chemical calculations along an air parcel trajectory ending in the region of the LOP on January 28. We now show those results in new Figures 11 and 12 (see below) and discuss them in the final paragraph of revised Section 4. Original figures 11-13 are now figures 13-15.*

[Figure]

Figure 11—Trajectory of air parcel that ends on January 28 at the location of the LOP. Numbers on the trajectory denote the date, beginning with January 15; the parcel is equatorward of 30°N on January 16-18.

[Figure]

Figure 12—Air parcel history of the changes in its (top left) pressure, (top right) ozone, (bottom left) NO₂, and (bottom right) HNO₃.

[Figure]

Figure S5—SH temperature (left) and ozone (right) at 0.032 hPa on January 15. Blue asterisk denotes satellite position (-75°S, 291°E); red asterisk denotes tangent point (-45°S, 294°E). LIMS view is from satellite to tangent point along temperature gradient maxima. Day/night terminator is near 60°S.